# Time-resolved studies define the nature of toxic IAPP intermediates, providing insight for anti-amyloidosis therapeutics

Andisheh Abedini[1]*, Annette Plesner[2†], Ping Cao[3], Zachary Ridgway[3], Jinghua Zhang[1], Ling-Hsien Tu[3], Chris T Middleton[4‡], Brian Chao[1], Daniel J Sartori[1], Fanling Meng[3], Hui Wang[3], Amy G Wong[3], Martin T Zanni[4], C Bruce Verchere[2], Daniel P Raleigh[3]*, Ann Marie Schmidt[1]*

[1]Diabetes Research Program, Division of Endocrinology, Diabetes and Metabolism, Department of Medicine, New York University School of Medicine, New York, United States; [2]Child and Family Research Institute, Department of Pathology and Laboratory Medicine and Department of Surgery, University of British Columbia, Vancouver, Canada; [3]Department of Chemistry, Stony Brook University, Stony Brook, United States; [4]Department of Chemistry, University of Wisconsin-Madison, Madison, United States

*For correspondence: andisheh. abedini@nyumc.org (AA); Daniel. Raleigh@stonybrook.edu (DPR); Annmarie.Schmidt@nyumc.org (AMS)

Present address: †Novo Nordisk, Bagsvaerd, Denmark; ‡Phase Tech spectroscopy,Inc, Madison

Competing interests: The authors declare that no competing interests exist.

**Abstract** Islet amyloidosis by IAPP contributes to pancreatic β-cell death in diabetes, but the nature of toxic IAPP species remains elusive. Using concurrent time-resolved biophysical and biological measurements, we define the toxic species produced during IAPP amyloid formation and link their properties to induction of rat INS-1 β-cell and murine islet toxicity. These globally flexible, low order oligomers upregulate pro-inflammatory markers and induce reactive oxygen species. They do not bind 1-anilnonaphthalene-8-sulphonic acid and lack extensive β-sheet structure. Aromatic interactions modulate, but are not required for toxicity. Not all IAPP oligomers are toxic; toxicity depends on their partially structured conformational states. Some anti-amyloid agents paradoxically prolong cytotoxicity by prolonging the lifetime of the toxic species. The data highlight the distinguishing properties of toxic IAPP oligomers and the common features that they share with toxic species reported for other amyloidogenic polypeptides, providing information for rational drug design to treat IAPP induced β-cell death.

## Introduction

The pathophysiological aggregation of polypeptides and proteins plays a key role in a wide range of protein misfolding diseases, including type 2 diabetes (T2D), Alzheimer's disease (AD) and systemic amyloidosis. Pancreatic islet amyloidosis by the neuropancreatic hormone, human islet amyloid polypeptide (h-IAPP, also known as amylin) contributes to β-cell death, progression of T2D, islet transplant failure, as well as cardiovascular complications (*Figure 1*) (*Potter et al., 2010*; *Ashcroft and Rorsman, 2012*; *Westermark et al., 2008*; *Despa et al., 2012*; *Abedini and Schmidt, 2013*; *Cao et al., 2013a*). Relatively little is known about the molecular properties that define the toxic species produced during amyloid formation (*Abedini and Schmidt, 2013*; *Cao et al., 2013a*; *Westermark et al., 1987*; *Cooper et al., 1987*; *Campioni et al., 2010*; *Chiti and Dobson, 2006*; *Johnson et al., 2012*; *Eisenberg and Jucker, 2012*; *Blancas-Mejía and Ramirez-Alvarado, 2013*), particularly in islet amyloidosis.

The physio-chemical properties of the toxic species produced during islet amyloidosis are not defined and there are no therapies for this pathology, in large part because of our limited

understanding of the molecular nature of the toxic species (*Abedini and Schmidt, 2013*; *Cao et al., 2013a*; *Gurlo et al., 2010*; *Hull et al., 2009*; *Janson et al., 1999*; *Masters et al., 2010*; *Park et al., 2012*; *Zhang et al., 2003*; *Zraika et al., 2009*; *Cooper et al., 2010*). It has been widely proposed that toxic oligomers produced by disparate proteins share many similar features (*Bolognesi et al., 2010*; *Chen et al., 2013*; *Chimon et al., 2007*; *Glabe, 2008*; *Kim et al., 2009*; *Laganowsky et al., 2012*; *Mannini et al., 2014*; *Bucciantini et al., 2002*). However, it is not known if IAPP oligomers are similar to toxic oligomers formed by other proteins; nor is it clear how oligomers formed by different proteins vary or how structured they are (*Chimon et al., 2007*; *Glabe, 2008*; *Lendel et al., 2014*; *Sandberg et al., 2010*). Here we use a multi-disciplinary approach to simultaneously monitor the real-time kinetics of IAPP toxicity and amyloid formation in solution, and measure the biochemical and physio-chemical properties of toxic and non-toxic IAPP species transiently produced over the course of aggregation, thereby linking specific molecular properties of amyloidogenic IAPP species to induction of β-cell death. The results provide important information about the nature of toxic IAPP oligomers, their unique properties and the common features they share with toxic entities produced in other amyloidosis diseases.

Mature, post-translationally modified h-IAPP (*Figure 1A*) is co-stored with insulin in the β-cell insulin secretory granules (~500 µM to low mM concentration range) and is co-secreted with insulin into the extracellular space within the pancreatic islets, where it then diffuses into blood vessels and enters the circulation (pM concentrations). The polypeptide plays an adaptive role in metabolism and glucose homeostasis, but in metabolic disease, h-IAPP forms pancreatic islet amyloid fibrils by an unknown mechanism (*Abedini and Schmidt, 2013*; *Cao et al., 2013a*; *Westermark et al., 2011*). The initiation site of islet amyloid formation is not known. Existing data indicate that both extracellular and intracellular h-IAPP oligomers contribute to islet β-cell toxicity. Histological studies show that amyloid deposits associated with T2D are extracellular (*Westermark et al., 2011*). Rodent IAPP is not toxic and does not form amyloid (*Westermark et al., 2011*), however, studies in transgenic rodent models that over-express h-IAPP and modulate the normal h-IAPP to insulin ratio suggest that islet amyloidosis may also have an intracellular origin. Intracellular aggregation of h-IAPP in these animal models suggests that defects in autophagy and/or endoplasmic reticulum (ER) stress play a role in toxicity; however, other reports argue that ER stress is not a significant contributor (*Hull et al., 2009*; *Huang et al., 2007*). There is strong evidence that extracellular oligomers induce cytotoxicity in vivo (*Westermark et al., 2011*; *Park et al., 2012*; *Zhang et al., 2003*; *Aston-Mourney et al., 2011*). Studies using a transgenic islet model that expresses human-relevant levels of h-IAPP demonstrate that h-IAPP secretion is required for amyloid formation and β-cell toxicity (*Aston-Mourney et al., 2011*). Receptor-mediated mechanisms of h-IAPP toxicity support a role for extracellular oligomers, as do studies showing that h-IAPP oligomers activate the inflammasome (*Johnson et al., 2012*; *Masters et al., 2010*; *Park et al., 2012*), and recent findings that extracellular h-IAPP oligomers can be transported into β-cells (*Trikha and Jeremic, 2013*; *Sheedy et al., 2013*). Thus, toxic h-IAPP oligomers can induce β-cell toxicity by both extra- and intracellular mechanisms. Here we focus on extracellular islet amyloidosis by h-IAPP.

## Results

### Toxic h-IAPP species are transient, pre-amyloid lag phase intermediates that upregulate oxidative stress, inflammation and apoptosis

Amyloid formation by h-IAPP, like that of other amyloidogenic proteins, comprises three distinct phenomenological phases: a lag, growth and saturation phase (*Figure 1B*). Little or no amyloid is formed in the lag phase and little is known about the nature of the species that populate this phase. Secondary nucleation leads to production of new fibrils, either by breakage of the small number of fibrils present or by templating new aggregates off the surface of existing fibrils. We developed time-resolved assays that allow concurrent biophysical, biochemical and biological characterization of the ensemble of species produced during IAPP amyloid formation (*Figure 2A*). Physiologically relevant solution conditions were found such that assembly of IAPP occurs on long time scales. The time scale is sufficiently long enough that the presence of toxic species can be detected indirectly by removing aliquots and applying them to cultured rat INS-1 β-cells or murine pancreatic islets. Stock solutions of h-IAPP, h-IAPP mutants and non-toxic, non-amyloidogenic rat IAPP (r-IAPP) were

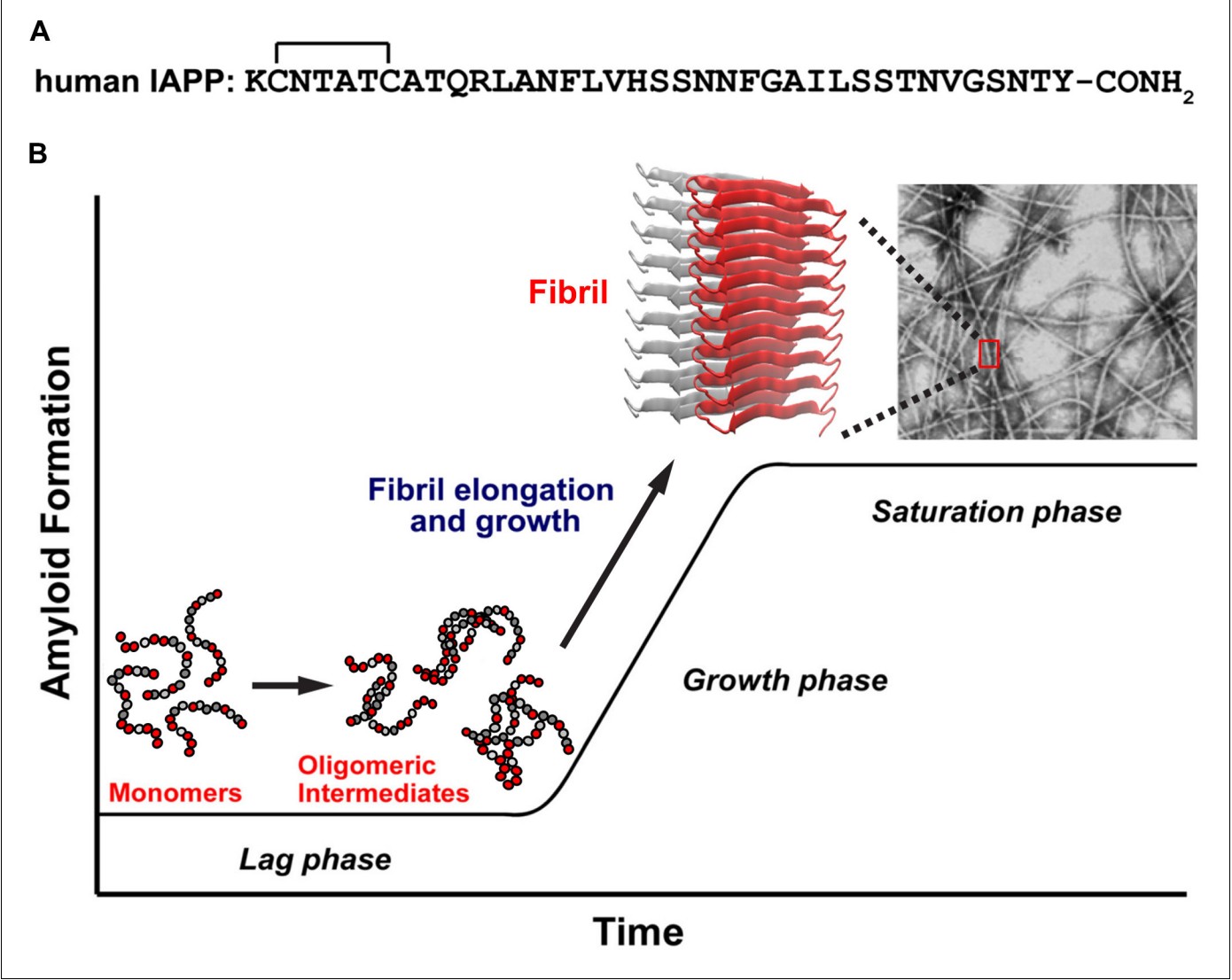

**Figure 1.** A schematic diagram of the process of amyloid formation by h-IAPP. (**A**) Amino acid sequence of wild-type h-IAPP. The mature, bioactive form of the polypeptide has an amidated C-terminus and a disulfide bridge indicated by the bracket between Cys-2 and Cys-7. (**B**) Schematic diagram illustrating the kinetics of amyloid formation by h-IAPP. The ribbon diagram shown is derived from the IAPP model developed by Eisenberg and co-workers (*Wiltzius et al., 2009*).

prepared by dissolving the peptides in buffer (time-zero) and incubating them at 25°C (pH 7.4). Aliquots were removed at various time points over the course of aggregation and characterized by the amyloid sensitive dye thioflavin-T and by transmission electron microscopy (TEM); aliquots were also applied to cultured β-cells at the same time points. Addition of aliquots to the cells involves only a 30% dilution of the peptide stock solutions. Control experiments using photochemical induced cross-linking and thioflavin-T kinetic assays of amyloid formation in buffer at 25°C reveal that this modest dilution does not significantly alter the distribution of oligomers, nor does it significantly alter the time course of amyloid formation. The same dilution into cell culture medium at 37°C has no significant effect on the time course (*Figure 2—figure supplements 1* and *2*). Toxicity was assessed by measuring loss in cellular metabolic function, detected by Alamar Blue reduction assays; production of reactive oxygen species (ROS); upregulation of inflammatory markers; production of cleaved caspase-3; and by observed changes in cellular morphology by light microscopy. These real-time experiments probe kinetic species produced during the course of h-IAPP amyloid formation,

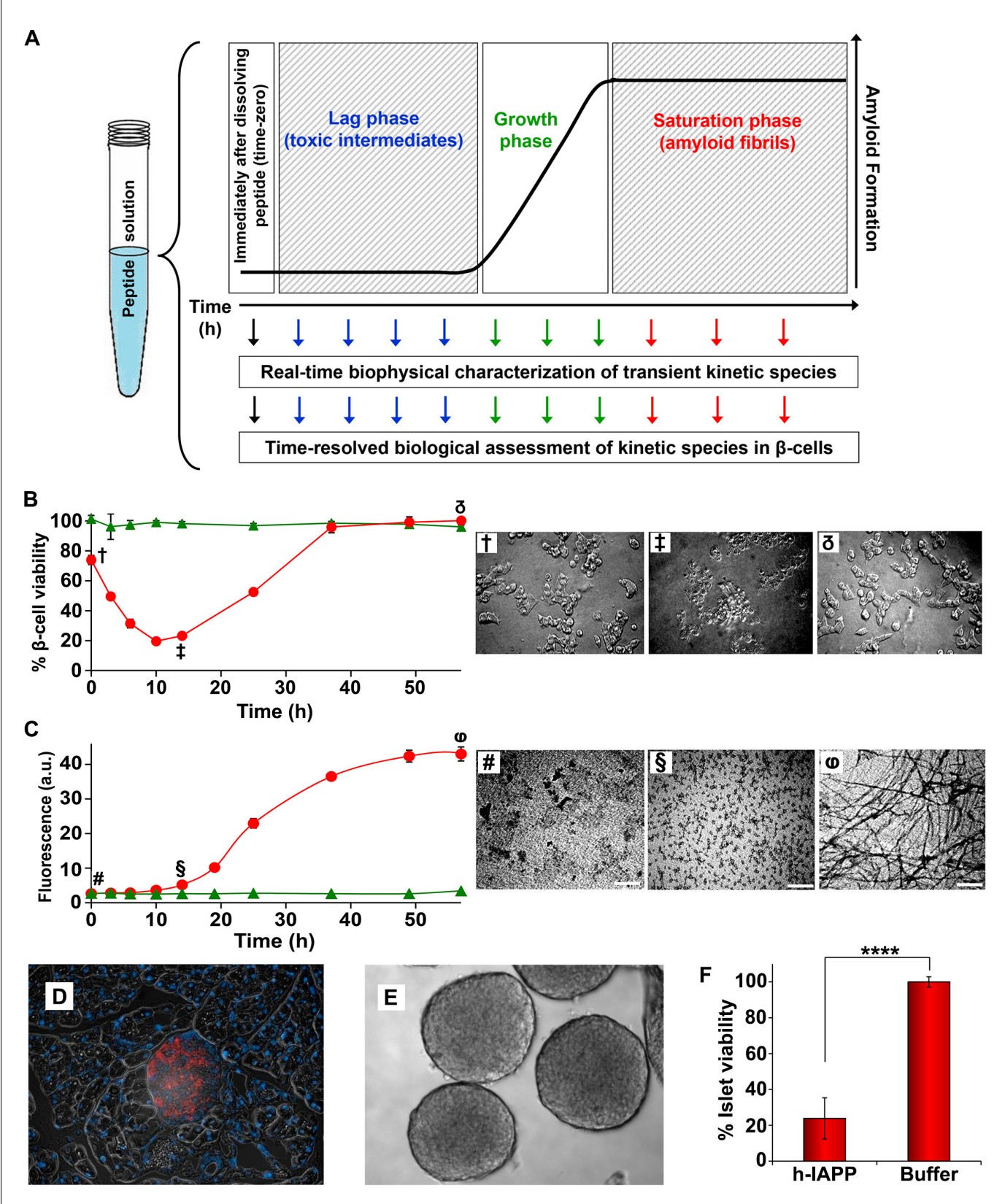

**Figure 2.** Toxic h-IAPP species are transiently populated lag phase intermediates. (**A**) A schematic diagram of the experimental design for the kinetic assays. Protein aggregation was initiated by dissolving amyloidogenic IAPP, non-amyloidogenic IAPP variants or r-IAPP in 20 mM Tris HCl buffer at pH

*Figure 2 continued on next page*

*Figure 2 continued*

7.4 (time-zero) followed by incubation. Aliquots were removed at designated time points over the course of amyloid formation (denoted by arrows) for concurrent biophysical characterization and biological assessment of transient kinetic species in rat INS-1 β-cells at the same time points, as described in the methods. (B) Time-resolved Alamar Blue reduction assays of INS-1 β-cells treated with h-IAPP (•) and r-IAPP (▲) at different time points during the course of aggregation. Light microscopy: (†) Viable β-cells after incubation with h-IAPP at time-zero; (‡) apoptotic β-cells shrink and detach from the cell culture substratum after incubation with lag phase intermediates; (δ) viable β-cells after incubation with amyloid fibrils. (C) Thioflavin-T monitored kinetics of amyloid formation: h-IAPP (•) and r-IAPP (▲). TEM images: (#) non-toxic h-IAPP at time-zero; (§) toxic pre-fibrillar lag phase intermediates; (φ) amyloid fibrils (Scale bars: 200 nm). (D) Immunofluorescence of a section of murine pancreas shows non-inflamed, insulin-positive islet: Sections of paraffin embedded pancreatic tissue were stained for insulin (red) indicative of β-cells, F4/80 (green) marker for macrophages indicative of inflammation and Dapi (blue) nuclear stain. (E) Light microscopy of hand purified pancreatic islets with intact mantels after isolation from wild-type mice. (F) Alamar Blue reduction assays show that h-IAPP lag phase intermediates are toxic to mouse pancreatic islets. Alamar Blue reduction in β-cell assays, thioflavin-T binding assays, light microscopy and TEM were conducted concurrently using aliquots from the same 20 μM peptide solutions. The peptide concentration after dilution into β-cell and islet assays was 14 μM. β-cell and islet viability is normalized to buffer treated cells or islets. Data represent mean ± SD of three to six replicate wells per condition and a minimum of three to ten replicate experiments per group (****$p<0.0001$). Some of the error bars in panels B and C are the same size or smaller than the symbols in the graphs. *Figure 2—figure supplements 1* and *2* provide control experiments for the biophysical and cellular assay conditions used in the studies.

The following figure supplements are available for figure 2:

**Figure supplement 1.** Dilution of h-IAPP by 30% does not change the distribution of the toxic oligomers.

**Figure supplement 2.** Dilution of h-IAPP by 30% into cell culture medium does not change the kinetics of amyloid formation.

and are fundamentally different from the common approach in which peptide is added to cells upon dissolution in cell culture medium and toxicity monitored after subsequent incubation times on cells. This experimental design also differs from studies that attempt to trap non-amyloidogenic oligomers using surfaces such as gold particles, detergents or micelles. It is not known how surface-trapping techniques affect the conformational properties of oligomers (*Bram et al., 2014*; *Kayed, 2003*). The experiments reported here provide critical information about toxic, amyloidogenic IAPP oligomers in solution.

h-IAPP toxicity to β-cells is observed to be time-dependent; amyloid fibrils are not toxic, but species populated in the lag phase are. Toxicity decreases in the growth phase and disappears in the saturation phase, directly indicating that the toxic species are transient lag phase intermediates (*Figure 2B and C*). Thioflavin-T binding assays and TEM studies confirm that the toxic intermediates are pre-fibrillar in nature. Aliquots of h-IAPP lag phase species appear to be amorphous and deposit on TEM grids as small spherical aggregates of various size, while species in the saturation phase exhibit long, unbranched amyloid fibril morphology (*Figure 2C*). We conducted additional biological experiments to determine whether h-IAPP lag phase intermediates produced in vitro are also toxic to pancreatic islets. We isolated and hand purified pancreatic islets from wild-type mice, confirmed the health and integrity of these organelles via immunofluorescence and light microscopy, and carried out ex vivo islet viability assays after incubation of the islets with either toxic h-IAPP lag phase intermediates or buffer control. The data provide direct evidence that the lag phase intermediates are toxic to cells in tissue. These results are consistent with our cellular studies and support our conclusion that h-IAPP lag phase intermediates are toxic to insulin producing pancreatic β-cells and primary islets (*Figure 2D,E and F*).

Cellular stress and inflammation have been implicated in h-IAPP induced β-cell toxicity in vitro, in mouse models of metabolic disease and in human T2D (*Westermark et al., 2011*; *Masters et al., 2010*; *Zraika et al., 2009*; *Janciauskiene and Ahrén, 2000*; *Konarkowska et al., 2005*; *Sakuraba et al., 2002*). If the lag phase intermediates identified here are toxic species then they should upregulate pro-inflammatory mediators and the production of ROS. This is exactly what was observed. Along with a decrease in β-cell viability, h-IAPP lag phase intermediates also induce an increase in *Ccl2* and *Il1b* mRNA expression, an increase in ROS production, upregulation of NADPH oxidase 1 (NOX1) protein expression, and an increase in cleaved caspase-3 production, consistent with h-IAPP induced β-cell stress, inflammation and apoptosis (*Figure 3A–D* and *Figure 3—figure supplements 1* and *2*). No significant upregulation of cytokines, ROS or cleaved caspase-3 production is induced by time-zero species or by h-IAPP amyloid fibrils, indicating that pro-inflammatory

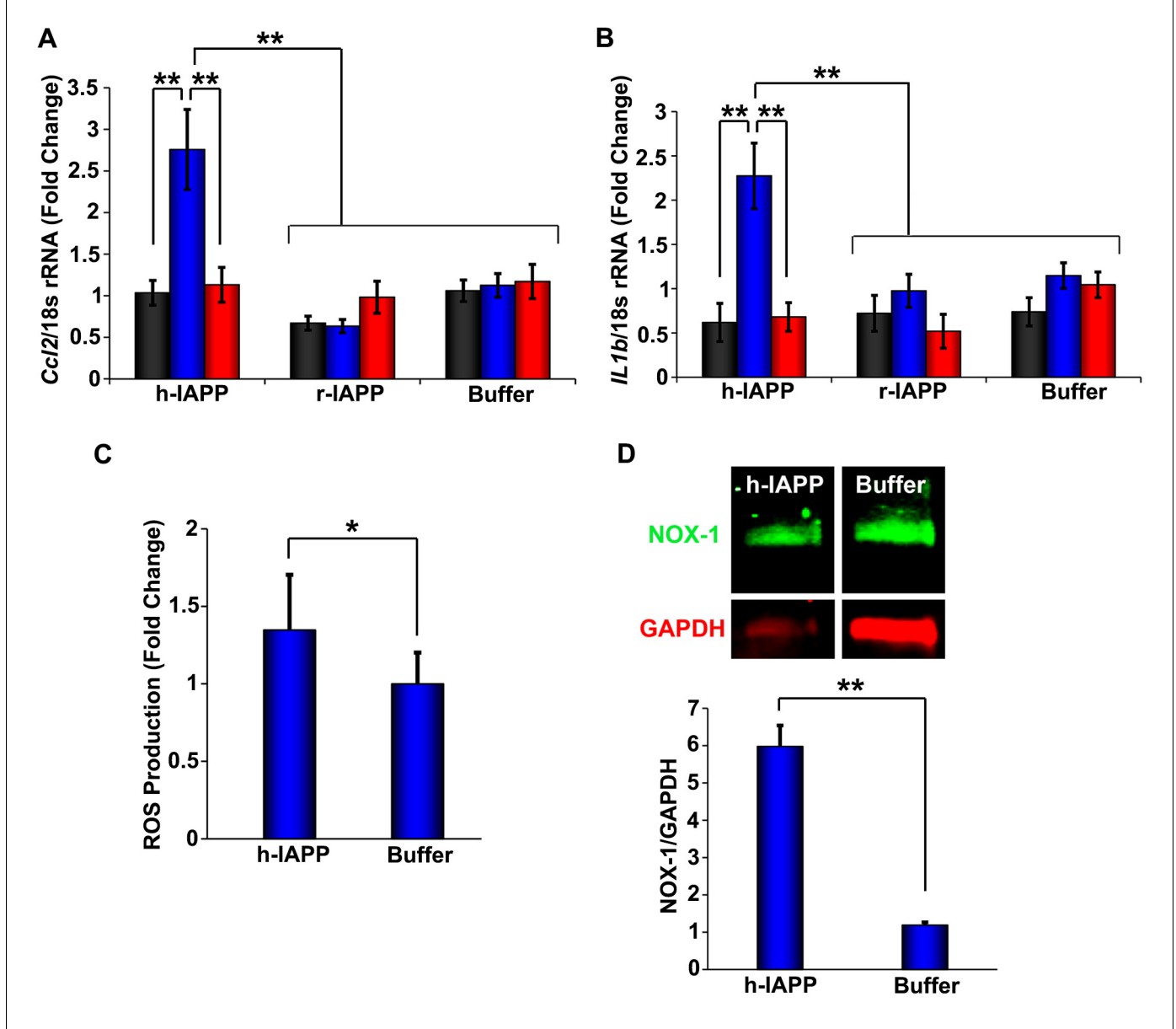

**Figure 3.** h-IAPP lag phase intermediates upregulate pro-inflammatory cytokines and oxidative stress. (A and B) qPCR studies of INS-1 β-cells treated with h-IAPP: Lag phase intermediates (blue) upregulate (A) *Ccl2* and (B) *Il1b*, but time-zero species (black), amyloid fibrils (red) and r-IAPP at the same time points do not. The peptide concentration after dilution into β-cell assays was 14 μM. (C) DHE-fluorescence assays of β-cells treated with h-IAPP lag phase intermediates show significant h-IAPP induced ROS production compared to control cells. (D) Western blot studies show upregulation of NOX1 in β-cells treated with h-IAPP lag phase intermediates compared to buffer treated cells. Data represent mean ± SD (DHE studies) and mean ± SEM (qPCR) of three to six replicate wells per condition and a minimum of three to ten replicate experiments per group (*p<0.05; **p<0.01). *Figure 3— figure supplements 1* and *2* provide additional concurrent experiments showing that h-IAPP induced loss in β-cell viability is accompanied by ROS and cleaved caspase-3 production.

The following figure supplements are available for figure 3:

**Figure supplement 1.** h-IAPP lag phase intermediates induce ROS production in INS-1 β-cells.

**Figure supplement 2.** Toxic h-IAPP lag phase intermediates induce β-cell apoptosis, but freshly dissolved h-IAPP (time-zero) and amyloid fibrils do not.

cellular responses are triggered specifically by pre-fibrillar lag phase intermediates. No toxicity or cytokine production is observed when non-amyloidogenic r-IAPP is added to cultured β-cells at any time point, consistent with previous reports (**Westermark et al., 2011**).

## Toxic h-IAPP lag phase intermediates are soluble, low order oligomers

Our ability to monitor toxicity in a time-resolved fashion allows us to characterize the physio-chemical properties of the toxic intermediates under well-defined conditions. Ultracentrifugation studies demonstrate that h-IAPP toxic species are soluble. Samples of toxic h-IAPP intermediates and amyloid fibrils were pelleted at 20,000 $g$ for 20 min and the soluble peptide remaining in the supernatant was measured. Control experiments confirm that r-IAPP is soluble under these conditions. At least 88% of h-IAPP is pelleted in the sample of fibrils, even at these low $g$-forces, while 94% of the peptide in the sample of toxic lag phase intermediates remains in the supernatant. TEM and thioflavin-T binding assays confirm the absence of amyloid in the supernatant of ultracentrifuged samples of toxic intermediates, and the presence of amyloid in the pellet obtained from samples of h-IAPP fibrils (**Figure 4A–D**). The supernatant of intermediates is toxic to β-cells, while the resuspended pellet from samples of amyloid fibrils is not, verifying that cytotoxic entities reside in the soluble phase and are not high molecular weight species (**Figure 4E**). Characterization of the ensemble of h-IAPP lag phase intermediates by circular dichroism (CD) reveals partial apparent helical structure. Positive signal is observed below 190 nm and a minima at 208 nm (**Figure 4F**). A second broad minima centered at 220 nm is also detected with a mean residue ellipticity on the order of $-6000$ (deg-cm2/dmol), consistent with transiently populated partial α-helical structure (**Manning and Woody, 1991**). However, helical and β-sheet CD signatures overlap in this region of the spectrum. Thus, the broad signal in this region may also include contributions from the presence of some β-sheet structure. Two dimensional infrared (2D IR) studies, described below, indicate that the overall level of β-structure is modest. With further incubation, the CD spectrum of h-IAPP changes and eventually converts into a spectrum indicative of β-structure (**Figure 4—figure supplement 1**). Aliquots of the supernatant from samples of toxic intermediates were characterized by CD, both before and after ultracentrifugation (**Figure 4F**). The spectra are superimposable, confirming that the observed CD signal reflects the peptide in the soluble fraction and demonstrates that the overall conformation of the ensemble of oligomeric intermediates in solution remains the same after ultracentrifugation.

We next sought to determine the approximate distribution of oligomeric species present in toxic h-IAPP solutions. Aliquots of toxic lag phase intermediates were trapped by in situ photochemical induced cross-linking and examined by SDS-PAGE, allowing differentiation between monomers and different size oligomers in solution. The in situ approach avoids concerns of structural perturbations induced by attaching photoactive groups and has been successfully used to study Aβ (**Bitan and Teplow, 2004**; **Bitan et al., 2001**). The data reveal a distribution of oligomers ranging from monomers to hexamers at time points of toxicity, confirming that the ensemble of toxic h-IAPP lag phase intermediates are soluble, low order oligomers (**Figure 4G and H**). Control studies show that the observed distribution is not an artifact of the irradiation time used for photochemical cross-linking (**Figure 4—figure supplement 2**). The dead time of the measurement (the time before the first measurement) is on the order of 10 min; a distribution of oligomers ranging from monomers to hexamers is populated within that time frame and the relative populations are similar to those detected later in the lag phase (**Figure 4G and H**, **Figure 4—figure supplement 3**). The data are consistent with independent ion mobility mass spectroscopy studies that report that a distribution of h-IAPP monomers to hexamers form within 2 min of initiating amyloid formation, and that the distribution is present later in the lag phase (**Young et al., 2014**). The rapid formation of oligomers and their persistence through the lag phase is consistent with recently proposed models of h-IAPP amyloid formation that posit that the lag phase could be controlled by a significant structural rearrangement within an oligomeric nucleus that involves crossing a high free energy barrier (**Buchanan et al., 2013**). Analysis of apparent relative populations of h-IAPP toxic species, deduced from the gel, indicate that dimers, trimers and tetramers are the most populated species. Monomeric proteins can be cross-linked by diffusion and collision of the photochemically modified monomers and it is important to show that the observed distribution differs from that expected for a monomeric protein (**Bitan and Teplow, 2004**). Thus, we employed a variant of the villin headpiece helical subdomain (HP35*), which is a soluble non-amyloidogenic protein of similar size to IAPP, as a control. The wild-

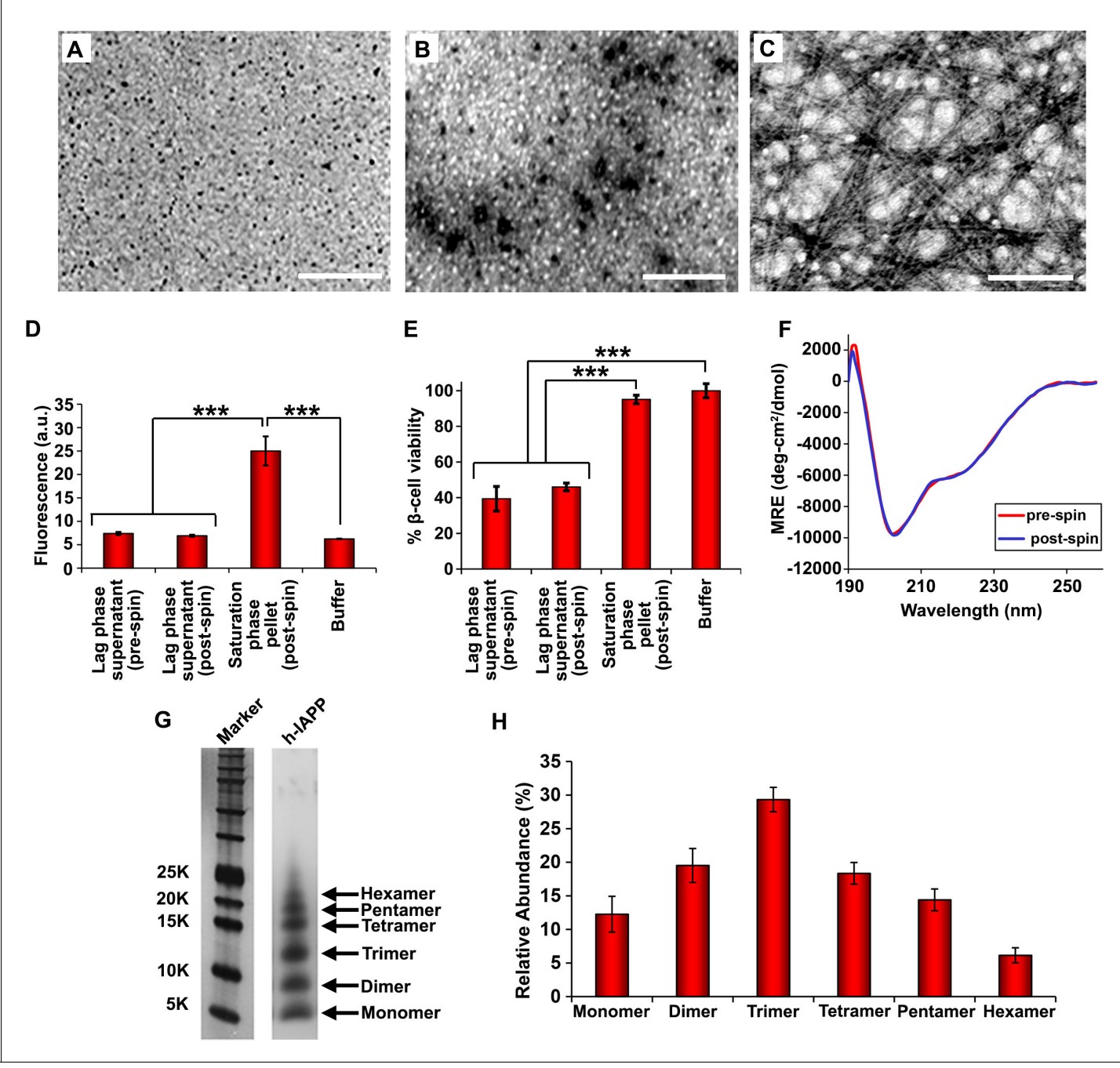

**Figure 4.** Toxic h-IAPP lag phase intermediates are soluble, low order oligomers with partial apparent α-helical structure. (A–C) TEM images: (A) r-IAPP, (B) supernatant of ultracentrifuged solution of h-IAPP lag phase intermediates produced after 10 h of incubation, and (C) resuspended pellet of ultracentrifuged solution of h-IAPP amyloid fibrils produced after 70 h of incubation (Scale bars: 200 nm). (D) Thioflavin-T binding assays confirm absence of amyloid fibrils in solutions of h-IAPP lag phase intermediates before ultracentrifugation and in the ultracentrifuged solutions of h-IAPP lag phase intermediates shown in panel B; they also confirm the presence of amyloid in the resuspended pellets of ultracentrifuged saturation phase solutions shown in panel C. The buffer control sample contains free thioflavin-T and the resulting fluorescence from this control solution is similar to that of free thioflavin-T solution by itself, and is thus the baseline. (E) Alamar Blue reduction assays show that the supernatant of samples of lag phase intermediates are toxic before and after ultracentrifugation, while the resuspended pellet of ultracentrifuged saturation phase samples are not toxic. (F) CD spectra of toxic lag phase intermediates before (red) and after (blue) ultracentrifugation. Data are plotted as mean residue ellipticity. (G) Representative SDS-PAGE of photochemically cross-linked toxic h-IAPP lag phase intermediates: lane-1, markers (molecular weight: KDaltons); lane-2, h-IAPP. (H) Quantitative analysis of the gels shown in panel G show a distribution of low order oligomers at time points of toxicity ranging from monomers to hexamers. Samples assessed in panels B–H were ultracentrifuged at 20,000 g for 20 min. Samples assessed in panels B–F used aliquots

*Figure 4 continued on next page*

*Figure 4 continued*

from the same peptide solutions. h-IAPP solutions contained 20 μM peptide. The peptide concentration after dilution into β-cell assays was 14 μM. Data represent mean ± SD of three to six replicate wells per condition and a minimum of three to nine replicate experiments per group (***p<0.001).

*Figure 4—figure supplements 1–5* provide additional biophysical characterization data for h-IAPP and a control peptide.

The following figure supplements are available for figure 4:

**Figure supplement 1.** Time-dependent Far UV CD data of h-IAPP.

**Figure supplement 2.** The detection of monomers through hexamers is not a consequence of the choice of irradiation time.

**Figure supplement 3.** The distribution of photochemically cross-linked oligomers detected for h-IAPP at 'time-zero' is similar to those detected for toxic h-IAPP lag phase intermediates.

**Figure supplement 4.** The distribution of photochemically cross-linked oligomers detected for h-IAPP is different from that observed for a monomeric protein of similar size.

**Figure supplement 5.** The distribution of photochemically cross-linked oligomers detected for solutions of non-toxic h-IAPP fibrils is significantly different than for toxic h-IAPP lag phase intermediates.

type subdomain contains a single Trp and a Met, but no Tyr. We replaced Trp with Tyr, and Met with nor-leucine to ensure that the control peptide contains the same photochemically active residues as h-IAPP. Quantitative analysis of the silver stained gels reveal that the distribution of cross-linked h-IAPP species is significantly different from that expected for a monomeric protein (*Figure 4H* and *Figure 4—figure supplement 4*). We also compared the observed oligomer distributions to those predicted by Teplow and coworkers for diffusing monomers of molecular weight 4 KDaltons (*Bitan et al., 2001*). That analysis considered the case of spherical monomers which do not interact except by diffusion with random elastic collisions. For low efficiency cross-linking, the model predicts that the most populated species is the monomer, and an approximately exponential decrease in intensity of high order species is predicted. Medium efficiency cross-linking still leads to the monomer being the dominate species, but to a shallower exponential decay in the relative populations. Both cases clearly differ from that observed for h-IAPP. High efficiency cross-linking is predicted to lead to further consumption of monomers and a shift in the maximum to dimers with the predicted monomer and dimer populations being noticeably higher than the predicted trimer, tetramer and pentamer populations. Again, this distribution is fundamentally different from that observed for h-IAPP lag phase species, where trimers are the most highly populated species and the population of pentamers is comparable to the population of monomers. The pattern of cross-linking observed for h-IAPP lag phase species is also very different than observed if pre-formed amyloid fibrils are cross-linked. h-IAPP was allowed to form fibrils and then the samples were centrifuged. No cross-linked h-IAPP oligomers were detected in the supernatant. Re-solubilization of the cross-linked fibrils revealed that the dominant species were monomers with some dimer present (*Figure 4—figure supplement 5*). The various control experiments together with comparison to independent mass spectrometry studies confirm that the observation of lag phase h-IAPP oligomers is robust.

## Not all IAPP oligomers are toxic

IAPP is expressed by all mammals examined to date; the amino acid sequences are ~80% conserved between species, however not all IAPP sequences are toxic or form amyloid in vivo (*Figure 5—figure supplement 1*) (*Betsholtz et al., 1989*; *Cao et al., 2013b*; *Westermark et al., 1990*). We wondered if non-amyloidogenic, non-toxic variants of IAPP oligomerize, and if so, whether the size distribution and/or the structure of the oligomers produced were significantly different. r-IAPP is non-toxic and non-amyloidogenic in vivo and is widely used as a negative control in biological and biophysical/biochemical studies of h-IAPP (*Westermark et al., 2011*). However, it aggregates and forms oligomers. To further validate the use of r-IAPP as a negative control, we carried out dose-response experiments to test the effect of incubating INS-1 β-cells with up to 6-fold higher

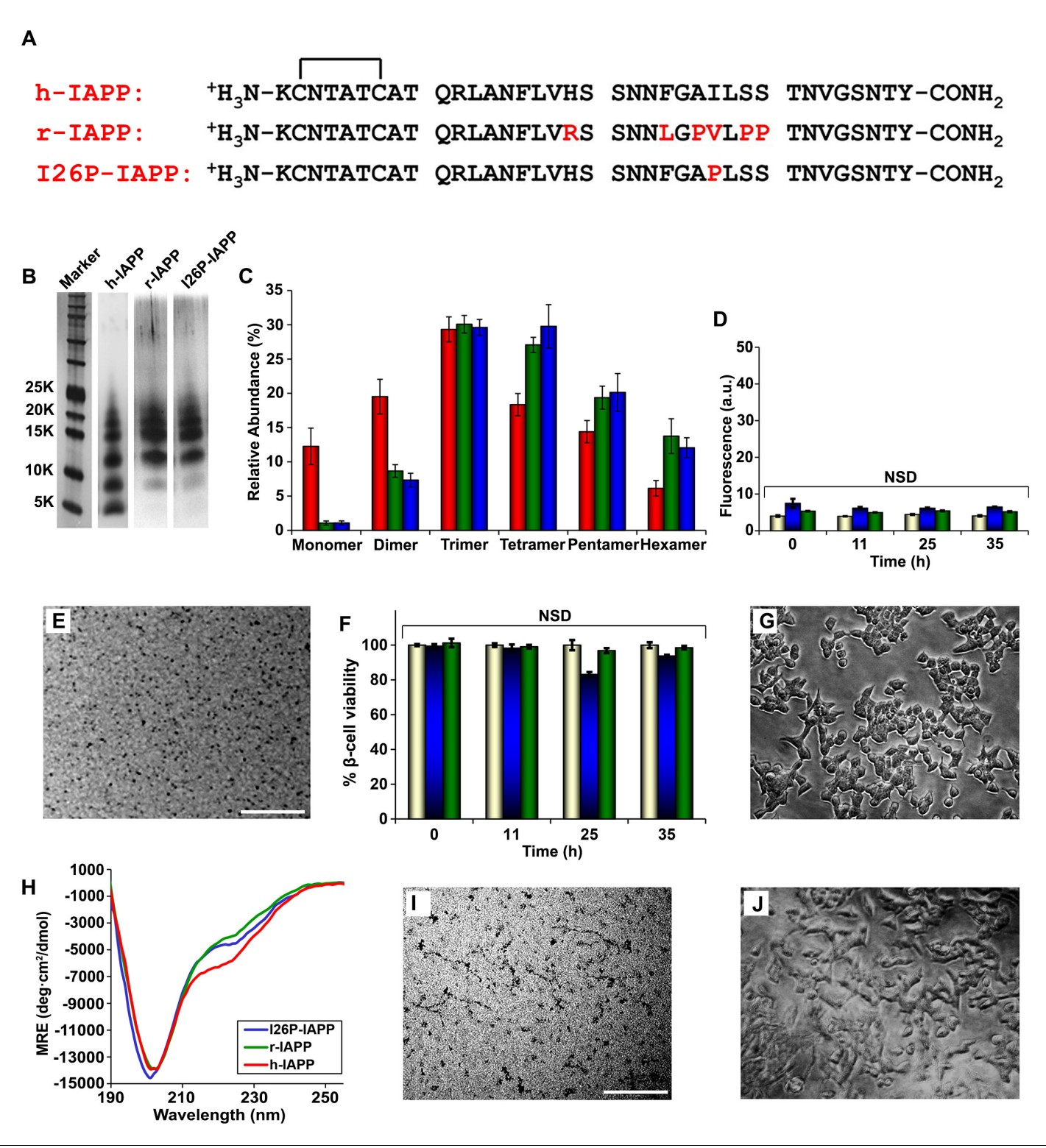

**Figure 5.** r-IAPP and I26P-IAPP form oligomers that are similar in size to those formed by h-IAPP, but do not form amyloid under the conditions of these studies, and are not toxic. (**A**) Primary sequence of h-IAPP, r-IAPP and I26P-IAPP. Mature polypeptides contain a disulfide between Cys2 and Cys7, indicated by brackets, and an amidated C-terminus. Amino acid positions that differ from h-IAPP are indicated in red. (**B**) Representative SDS-PAGE of photochemically cross-linked non-toxic r-IAPP and I26P-IAPP show the presence of low order oligomers. Peptide solutions were incubated at 25°C until time points corresponding to h-IAPP toxicity and ultracentrifuged for 20,000 *g* for 20 min. Aliquots of the supernatants were removed and

*Figure 5 continued on next page*

*Figure 5 continued*

irradiated for 10 s for cross-linking, and analyzed using silver staining. Lane-1, markers (molecular weight: KDaltons); lane-2, h-IAPP; lane 3, r-IAPP; and lane 4, I26P-IAPP. (C) Quantitative analysis of the gels shown in panel B reveal a distribution of low order oligomers ranging up to hexamers: h-IAPP (red); r-IAPP (green); and I26P-IAPP (blue). (D) Thioflavin-T binding assays of aliquots of buffer (gold), r-IAPP (green) or I26P-IAPP (blue) at different time points over the course of aggregation. (E) TEM image of r-IAPP oligomers after 14 h incubation (Scale bars: 200 nm). (F) Alamar Blue reduction assays of β-cells treated with buffer (gold), r-IAPP (green) or I26P-IAPP (blue) at different time points over the course of aggregation. (G) Light microscopy image of viable β-cells after treatment with r-IAPP aggregates shown in panel E. (H) CD spectra of I26P-IAPP (blue), r-IAPP (green) and h-IAPP (red). Data is plotted as mean residue ellipticity. (I) TEM image of I26P-IAPP oligomers after 14 h incubation. (J) Light microscopy image of viable β-cells treated with oligomers shown in panel I. IAPP solutions contained 20 μM peptide. The final peptide concentration after dilution into β-cell assays was 14 μM. Data represent mean ± SD of three to six replicate wells per condition and three replicate experiments per group (NSD: no significant difference; Scale bars: 200 nm). *Figure 5—figure supplements 1–7* provide additional biochemical information, biophysical characterization and toxicity experiments for the non-toxic, non-amyloidogenic r-IAPP and h-IAPP mutants.

The following figure supplements are available for figure 5:

**Figure supplement 1.** Primary sequences of IAPP from different species.

**Figure supplement 2.** Dose-response studies show r-IAPP is not toxic.

**Figure supplement 3.** The CD spectrum of r-IAPP reveals random coil conformation and is independent of concentration and time.

**Figure supplement 4.** Time-dependent far UV CD data of I26P-IAPP.

**Figure supplement 5.** A designed, non-toxic H18R, G24P, I26P triple mutant of h-IAPP (TM-IAPP) oligomerizes.

**Figure supplement 6.** Hydropathy plots for IAPP peptides.

**Figure supplement 7.** Average per residue hydrophobicity for different IAPP peptides.

concentrations (84 μM) of r-IAPP for up to 19-fold longer incubation times on cells (96 h) than used in the assays employed herein to assess h-IAPP toxicity. No detectable toxicity was observed for r-IAPP, even at these significantly higher concentrations (*Figure 5—figure supplement 2*). The r-IAPP sequence differs from the h-IAPP sequence at six positions and contains three Pro residues and a His-18 to Arg replacement (*Figure 5A*). Pro is a well-known breaker of secondary structure and substitution of His with Arg will increase the net charge of the peptide. Cross-linking studies (*Figure 5B*) reveal a broadly similar distribution of oligomers produced in solution by r-IAPP and h-IAPP, with detected species ranging from monomers to hexamers. There are some differences in the relative intensities of the different oligomeric states, but these are relatively modest and it is difficult to unambiguously deduce their significance. The important feature is that the rat polypeptide clearly oligomerizes and forms dimers to hexamers similar to that of the human peptide; yet, it is not toxic (*Figure 5C*). Characterization by thioflavin-T binding assays and TEM confirm that r-IAPP does not form amyloid fibrils during these experiments (*Figure 5D and E*). Independent ion mobility mass spectroscopy (IM-MS) studies have also shown that h-IAPP and r-IAPP form similar distributions of oligomers (*Young et al., 2014*). Cell viability assays carried out simultaneously with biophysical measurements using aliquots from the same stock solutions show that r-IAPP is not toxic at any time point under these conditions, even though it oligomerizes (*Figure 5F and G*). Additional studies show that r-IAPP is not toxic over a 28 day time course (data not shown). CD studies reveal that r-IAPP oligomers appear less structured than their h-IAPP counterparts, as indicated by a positive signal below 190 nm and less intense signal between 218 to 222 nm (*Figure 5H*) in the spectrum of lag phase oligomers. While the differences in the CD spectra of r-IAPP and h-IAPP are moderate, they are significant and reproducible using different preparations of both peptides and in independent experiments conducted by different investigators. In contrast to h-IAPP, the CD spectrum of non-amyloidogenic r-IAPP does not change with time and is independent of concentration over the range tested (*Figure 5—figure supplement 3*). To further test if toxicity is decoupled from general aggregation, we examined an I26P point mutant of h-IAPP (I26P-IAPP) (*Figure 5A*). We have previously shown that I26P-IAPP inhibits amyloid formation by h-IAPP and does not form amyloid by itself

under the conditions of our studies (*Abedini et al., 2007*; *Meng et al., 2010*). I26P-IAPP is similar to h-IAPP in hydrophobicity and has an identical net charge. Like r-IAPP, this mutant forms low order oligomers with an apparent size distribution that is similar to h-IAPP oligomers, as judged by photochemical induced cross-linking studies, but is non-amyloidogenic under these conditions over the 35 + h duration of these studies, as judged by thioflavin-T binding assays and TEM (*Figure 5D and I*). The CD spectrum of I26P-IAPP is similar to that of r-IAPP (*Figure 5H* and *Figure 5—figure supplement 4*). Cell viability assays carried out in parallel with biophysical studies show that I26P-IAPP is not toxic at any time point in our studies (*Figure 5F and J*). As an additional control, we analyzed a recently described non-toxic variant of h-IAPP (*Wang et al., 2014a*) (*Figure 5—figure supplement 5A*). The H18R, G24P, I26P triple mutant of h-IAPP (TM-IAPP) has been shown to be non-amyloidogenic and non-toxic (*Wang et al., 2014b*). Photochemical induced cross-linking and CD studies show that this variant also oligomerizes, even though it remains as random coil as judged by CD (*Figure 5—figure supplement 5B–D*). Thus, all three of the different non-toxic variants, which have similar hydrophobicity to h-IAPP, oligomerize (*Figure 5—figure supplements 6* and *7*). It is not possible to resolve the structural differences between the transiently populated ensemble of h-IAPP oligomers and the ensembles populated by r-IAPP, I26P-IAPP and TM-IAPP. However, the key point is that properties of the polypeptides beyond their ability to oligomerize are clearly important determinants of cellular toxicity. It is interesting to note that there are differences in the distribution of h-IAPP oligomers and the non-toxic r-IAPP and I26P-IAPP variants. Relatively more dimer is detected for h-IAPP compared to these two non-toxic variants and a reduction in the relative population of pentamers and hexamers is also detected. This may reflect actual differences in the distribution of oligomers in solution or it may include contributions from changes in cross-linking efficiency. The data cannot differentiate between the two potential explanations. The key feature is that these results decouple general aggregation and oligomer formation from toxic species formation, and suggest that the conformational properties of oligomers, and not their size, are important determinants of cellular toxicity.

## The ensemble of toxic h-IAPP oligomers contain modest overall β-sheet structure, in contrast to reports on toxic species derived from Aβ and other amyloidogenic proteins

We probed the conformational properties of toxic h-IAPP oligomers in more detail to determine how they compared with those reported for toxic species formed by other amyloidogenic proteins. Of particular interest is a comparison with the Aβ peptide of AD, given the similarity between the two polypeptides and recent studies that suggest a link between AD and T2D (*Yang and Song, 2013*). h-IAPP and Aβ40 have 25% amino acid identity and 50% similarity with segments believed to be important for the self-assembly of each peptide, h-IAPP (*Park et al., 2012*; *Zhang et al., 2003*; *Zraika et al., 2009*; *Cooper et al., 2010*; *Bolognesi et al., 2010*; *Chen et al., 2013*; *Chimon et al., 2007*; *Glabe, 2008*; *Kim et al., 2009*; *Laganowsky et al., 2012*) and Aβ40 (*Chimon et al., 2007*; *Glabe, 2008*; *Kim et al., 2009*; *Laganowsky et al., 2012*; *Mannini et al., 2014*; *Bucciantini et al., 2002*; *Lendel et al., 2014*) having high similarity (*Figure 6—figure supplement 1*). Aβ fibrils can seed amyloid formation by h-IAPP in vitro and in an animal model, and the two polypeptides interact in vitro (*Andreetto et al., 2010*; *O'Nuallain et al., 2004*; *Oskarsson et al., 2015*). Recent work has revealed significant levels of β-sheet structure in toxic oligomers from several proteins, including Aβ (*Chimon et al., 2007*; *Laganowsky et al., 2012*; *Lendel et al., 2014*; *Sandberg et al., 2010*; *Do et al., 2016*). We detected apparent partial helical structure in the ensemble of toxic h-IAPP lag phase oligomers by CD (*Figure 4E*, *5H* and *Figure 4—figure supplement 1*). Observation of partial helical structure is consistent with studies of truncated h-IAPP analogs fused to maltose binding protein; as well as studies of h-IAPP aromatic residue mutants, and NMR studies of soluble IAPP variants (*Wiltzius et al., 2009*; *Williamson et al., 2009*; *Tu and Raleigh, 2013*). It is also known that helical structure can be stabilized in h-IAPP by binding to negatively charged surfaces such as sulfated glycosaminoglycans or to vesicles containing significant amounts of anionic lipids (*Wiltzius et al., 2009*; *Williamson et al., 2009*; *Tu and Raleigh, 2013*; *Brender et al., 2012*; *Knight et al., 2006*; *Meng et al., 2007*). CD is well suited to probe helical structure, but is less sensitive to the details of β-sheet structure; individual β-sheets can exhibit significant differences in their CD signal. Thus, we applied newly developed 2D IR methods. 2D IR is a sensitive probe of β-sheet structure in aggregating systems (*Buchanan et al., 2013*; *Wang et al., 2011*; *Strasfeld et al., 2009*). The spectrum of

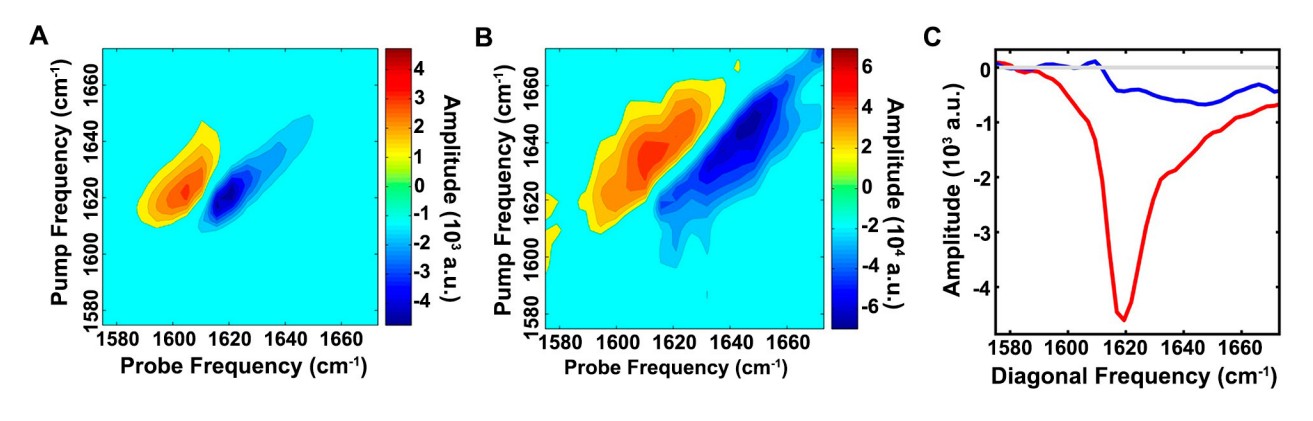

**Figure 6.** The ensemble of toxic h-IAPP oligomers contain only modest amounts of overall β-sheet structure. 2D IR spectra of h-IAPP: (**A**) Amyloid fibrils are rich in β-sheet structure, but (**B**) lag phase intermediates show no significant (<15%) β-sheet structure. The spectra in panels **A** and **B** are plotted on different intensity scales. (**C**) Comparison of the intensity of the diagonal slice in panels **A** and **B**: Intermediates (blue), fibrils (red) and zero baseline (grey). *Figure 6—figure supplement 1* provides sequence alignment analysis of h-IAPP and Aβ40.

The following figure supplement is available for figure 6:

**Figure supplement 1.** Sequence alignment of h-IAPP with Aβ40.

h-IAPP amyloid fibrils has significant intensity along the diagonal in the β-sheet region between 1615 and 1625 cm$^{-1}$ (*Figure 6A*) and is in good agreement with published spectra of h-IAPP amyloid fibrils. A flexible, partially structured intermediate will yield a significantly less intense 2D IR spectrum than a well-ordered β-structure, since the parallel β-sheet structure in amyloid fibrils leads to a large transition dipole. The spectrum of the intermediates is much less intense than the spectrum of the amyloid fibrils, indicating only modest levels of β-sheet structure (*Figure 6B*). Based on the relative areas in *Figure 6C*, the upper limit for the β-sheet content in the ensemble of lag phase intermediates populated under these conditions is estimated to be on the order of 15%. This does not preclude a high level of β-structure in a short segment of the protein. Recent isotope edited 2D IR studies using 10- to 20-fold higher h-IAPP concentrations than used herein (the isotope edited studies cannot currently be conducted at 40 μM peptide for technical reasons) suggest that h-IAPP forms an intermediate with well developed, parallel, in–register β-structure in the FGAIL region during amyloid formation under those conditions (*Buchanan et al., 2013*). A β-sheet of this size is fully consistent with the 2D IR data presented here.

## The ensemble of h-IAPP toxic oligomers are globally flexible, have solvated aromatic side chains and do not bind 1-anilnonaphthalene-8-sulphonic acid (ANS), bis-ANS or Nile Red

ANS, a dye that is widely employed in protein folding studies to detect exposed hydrophobic patches and molten globule states (*Figure 7A*), binds to toxic pre-amyloid oligomers formed by a range of other amyloidogenic proteins, including various oligomers formed by the Aβ peptide, lysozyme, the α-synuclein protein of Parkinson's disease, SH3 domains, HypF-N, bovine serum albumin, concanavalin and others (*Bolognesi et al., 2010*; *Mannini et al., 2014*; *Frare et al., 2009*; *Lorenzen et al., 2014*; *Bhattacharya et al., 2011*; *Fu et al., 2015*; *Ghosh et al., 2015*; *Paslawski et al., 2014*; *Vetri et al., 2013*). We tested if toxic h-IAPP lag phase intermediates bind ANS. No ANS binding is observed in the lag phase, but is observed during the growth phase of amyloid formation and in the saturation phase containing fibrils (*Figure 7B and C*). Thus, the properties of h-IAPP toxic oligomers are distinct from those recently described for certain other amyloidogenic proteins (*Bolognesi et al., 2010*; *Kim et al., 2009*; *Laganowsky et al., 2012*; *Mannini et al., 2014*; *Lendel et al., 2014*; *Sandberg et al., 2010*; *Frare et al., 2009*; *Lorenzen et al., 2014*; *Stroud et al., 2012*). 4,4′-Dianilino-1,1′-binaphthyl-5,5′-disulfonic acid (bis-ANS) has also been used to probe molten globule states, the formation of exposed hydrophobic patches, and, in limited

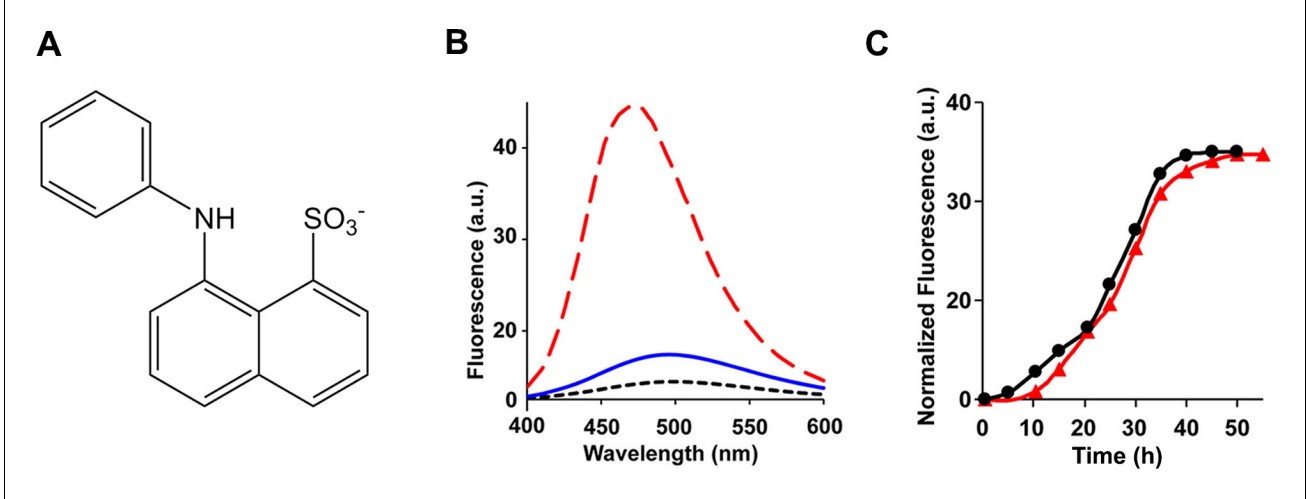

**Figure 7.** The ensemble of toxic h-IAPP oligomers do not bind ANS. (**A**) Structural model of an ANS molecule. (**B**) ANS fluorescence emission spectra of h-IAPP at time-zero (black, ····), lag phase intermediates (blue, —) and amyloid fibrils (red, - - - -). (**C**) Kinetic assays monitored by ANS binding (•) and thioflavin-T binding (▲) confirm that h-IAPP lag phase intermediates do not bind ANS. *Figure 7—figure supplements 1–14* provide additional dye-binding studies using bis-ANS and Nile Red, and biophysical characterization of the toxic h-IAPP lag phase intermediates.

The following figure supplements are available for figure 7:

**Figure supplement 1.** Bis-ANS and Nile Red do not bind to h-IAPP lag phase intermediates.

**Figure supplement 2.** Fluorescence detected thioflavin-T binding assay (black) showing the kinetics of amyloid formation by a solution of h-IAPP used in the proteolytic digestion studies presented in *Figure 7—figure supplements 3–14*.

**Figure supplement 3.** Characterization of h-IAPP time-zero species by MALDI-TOF MS.

**Figure supplement 4.** Characterization of h-IAPP early lag phase species by MALDI-TOF MS.

**Figure supplement 5.** Characterization of h-IAPP mid-lag phase species by MALDI-TOF MS.

**Figure supplement 6.** Characterization of h-IAPP amyloid fibrils by MALDI-TOF MS.

**Figure supplement 7.** Characterization of h-IAPP time-zero species by five minute Proteinase K digestion as monitored by MALDI-TOF MS.

**Figure supplement 8.** Characterization of h-IAPP early lag phase species by five minute Proteinase K digestion as monitored by MALDI-TOF MS.

**Figure supplement 9.** Characterization of h-IAPP mid-lag phase species by five minute Proteinase K digestion as monitored by MALDI-TOF MS.

**Figure supplement 10.** Characterization of h-IAPP amyloid fibrils by five minute Proteinase K digestion as monitored by MALDI-TOF MS.

**Figure supplement 11.** Characterization of h-IAPP time-zero species by forty minute Proteinase K digestion as monitored by MALDI-TOF MS.

**Figure supplement 12.** Characterization of h-IAPP early lag phase species by forty minute Proteinase K digestion as monitored by MALDI-TOF MS.

**Figure supplement 13.** Characterization of h-IAPP mid-lag phase species by forty minute Proteinase K digestion as monitored by MALDI-TOF MS.

**Figure supplement 14.** Characterization of h-IAPP amyloid fibrils by forty minute Proteinase K digestion as monitored by MALDI-TOF MS.

applications, amyloid formation (*Younan and Viles, 2015*; *Hawe et al., 2008*). Binding of bis-ANS to partially folded states often leads to a larger fluorescence change than ANS binding. We also tested the ability of h-IAPP toxic intermediates to bind bis-ANS. No binding to lag phase species is

observed, but the dye, like ANS, binds to h-IAPP amyloid fibrils (*Figure 7—figure supplement 1*). Nile Red, like ANS, is used as a fluorescent probe of hydrophobic protein surfaces and has been shown to bind to pre-amyloid oligomers formed by some amyloidogenic proteins, but recent studies, conducted under different conditions than employed in our work, show that it does not bind to h-IAPP lag phase intermediates (*Hawe et al., 2008*; *Jha et al., 2014*; *Krishnan et al., 2012*; *Sackett and Wolff, 1987*). We independently confirmed that it does not bind to the lag phase species populated in our experiments (*Figure 7—figure supplement 1*).

We examined the susceptibility of the oligomeric lag phase intermediates to proteolytic digestion by Proteinase K in order to further probe their structure and flexibility. h-IAPP monomers and lag phase intermediates are rapidly digested by the protease, while h-IAPP amyloid fibrils are not, even after 40 min of incubation (*Figure 7—figure supplement 2–14*). The results indicate that the intermediates are much less structured than the amyloid fibrils, and show that h-IAPP amyloid fibrils have similar anti-protease properties as amyloid fibrils derived from other proteins.

We next probed the solvent exposure of the three aromatic residues of h-IAPP using the non-genetically coded fluorescent amino acid, *p*-cyano-phenylalanine (*p*-cyanoPhe) (*Figure 8*). *p*-Cyano-Phe can be incorporated into proteins and used to follow amyloid formation (*Marek et al., 2010a*). Its fluorescence is high when the cyano-group is solvent exposed and hydrogen bonded, and low when it is not; the fluorescence is also quenched via FRET to Tyr with a Ro of 15Å. h-IAPP contains two Phe and one Tyr; thus, three analogs were prepared in which one aromatic residue was replaced by *p*-cyanoPhe at each position (*Figure 8A-D*). Fluorescence is high for unaggregated h-IAPP (time-zero) and is quenched in the amyloid fibrils. The fluorescence intensity of the lag phase intermediates is also high and shows only moderate differences from the value observed for each peptide at time-zero, but is much higher than the intensity observed from the amyloid fibrils (*Figure 8E*). The data indicate that Phe-15, Phe-23 and Tyr-37 are largely solvent exposed in the lag phase, and rule out a significant population of conformations in the ensemble in which the aromatic residues are buried, or in which the C-terminal Tyr forms persistent interactions with either Phe-15 or Phe-23. Again, these results are compatible with 2D IR studies undertaken at higher peptide concentrations, which postulate formation of β-sheet structure in the FGAIL region.

Collectively, the data show that the ensemble of toxic lag phase oligomers of h-IAPP are defined by the following characteristics: they are soluble, globally flexible, lack extensive β-sheet structure, and do not have persistent hydrophobic surface patches that allow ANS, bis-ANS or Nile Red binding. The data does not preclude short regions of well-ordered polypeptide and intermolecular hydrogen bonding provided such interactions do not lead to significant sequestering of the aromatic residues from solvent, development of ANS binding surfaces or significant protection of Proteinase K cleavage sites. This combination of properties indicates that toxic h-IAPP lag phase oligomers share similar features with those reported for a range of other amyloidogenic proteins, but are not identical to them (*Bolognesi et al., 2010*; *Kim et al., 2009*; *Laganowsky et al., 2012*; *Mannini et al., 2014*; *Lendel et al., 2014*; *Sandberg et al., 2010*; *Frare et al., 2009*; *Lorenzen et al., 2014*; *Stroud et al., 2012*; *Chen et al., 2015*). The distinct molecular features of toxic IAPP oligomers have important implications for rational design of drugs with improved specificity.

## Aromatic-aromatic and aromatic-hydrophobic interactions are not required for toxicity, but do contribute to toxicity

Aromatic contacts (π-π interactions) have been proposed to play an important role in amyloid formation. Although they are not required for h-IAPP amyloid formation, mutation of the three aromatic residues in h-IAPP to Leu (3xL-IAPP) slows the rate of amyloid formation (*Tu and Raleigh, 2013*; *Marek et al., 2007*; *Gazit, 2007*). Our *p*-cyanoPhe experiments show that there are no persistent interactions between F15 and Y37 or F23 and Y37 of h-IAPP, but the studies are less sensitive to formation of low levels (<5 to 10%) of conformers with F15/Y37 or F23/Y37 contacts, and do not probe potential interactions between F15 and F23. Consequently, we examined a triple mutant of h-IAPP: F15L, F23L, Y37L-IAPP (3xL-IAPP) that lacks aromatic residues to test whether or not aromatic π-π interactions or aromatic-hydrophobic interactions are required for toxicity (*Figure 9A*). Thioflavin-T assays and TEM measurements confirm that the triple mutant does form amyloid more slowly than the human peptide (*Figure 9B–D*). Cell viability studies show, that like h-IAPP, 3XL-IAPP exhibits time dependent toxicity; the amyloid fibrils produced by 3xL-IAPP are not toxic to β-cells, but

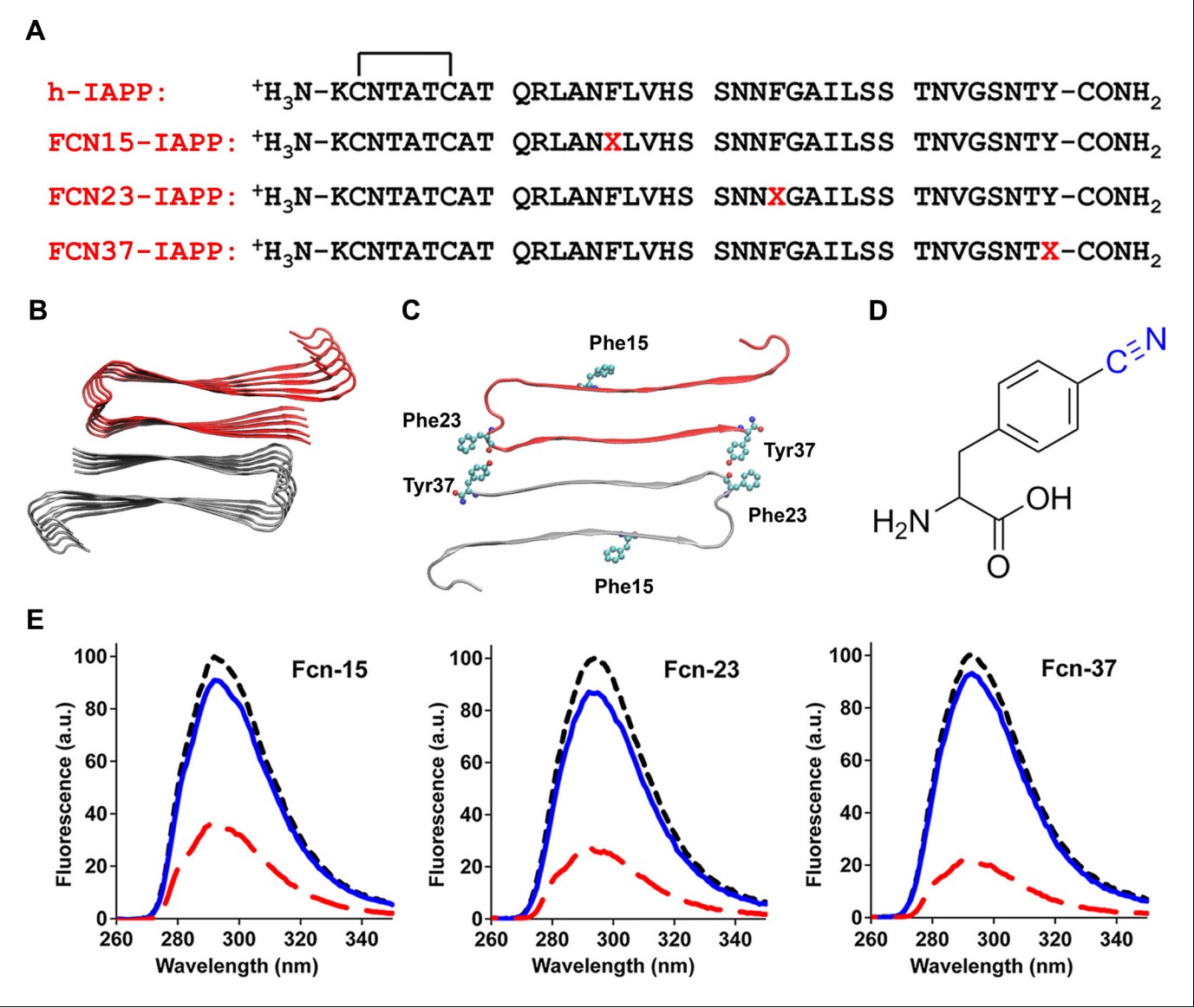

**Figure 8.** Aromatic residues in the ensemble of toxic h-IAPP oligomers are solvent exposed. (A) Primary sequences of h-IAPP and *p*-cyano-phenylalanine variants; red X=cyanophenylalanine. (B) A structural model of the h-IAPP amyloid fibril. (C) Location of aromatic residues in h-IAPP which are replaced with *p*-cyano-phenylalanine in the h-IAPP variants. (D) Structure of the unnatural amino acid *p*-cyano-phenylalanine. (E) *p*-Cyano-phenylalanine fluorescence emission spectra reveal that aromatic side chains are solvent exposed in time-zero species (black, ····) and lag phase intermediates (blue, —), but are buried in amyloid fibrils (red, - - - -).

species populated in the lag phase are, further confirming that toxicity resides with pre-amyloid intermediates (*Figure 9B–E*). Dose-response studies show that 3xL-IAPP evokes significantly lower levels of toxicity than h-IAPP. At their respective time points of maximum toxicity, 40 μM 3xL-IAPP reduced β-cell viability to 67%, while 20 μM h-IAPP reduced β-cell viability to 35% (*Figure 9F*). However, 3xL-IAPP is clearly still toxic, indicating that aromatic residues, and hence π-π interactions, are not an absolute requirement for h-IAPP toxicity, but do contribute to it.

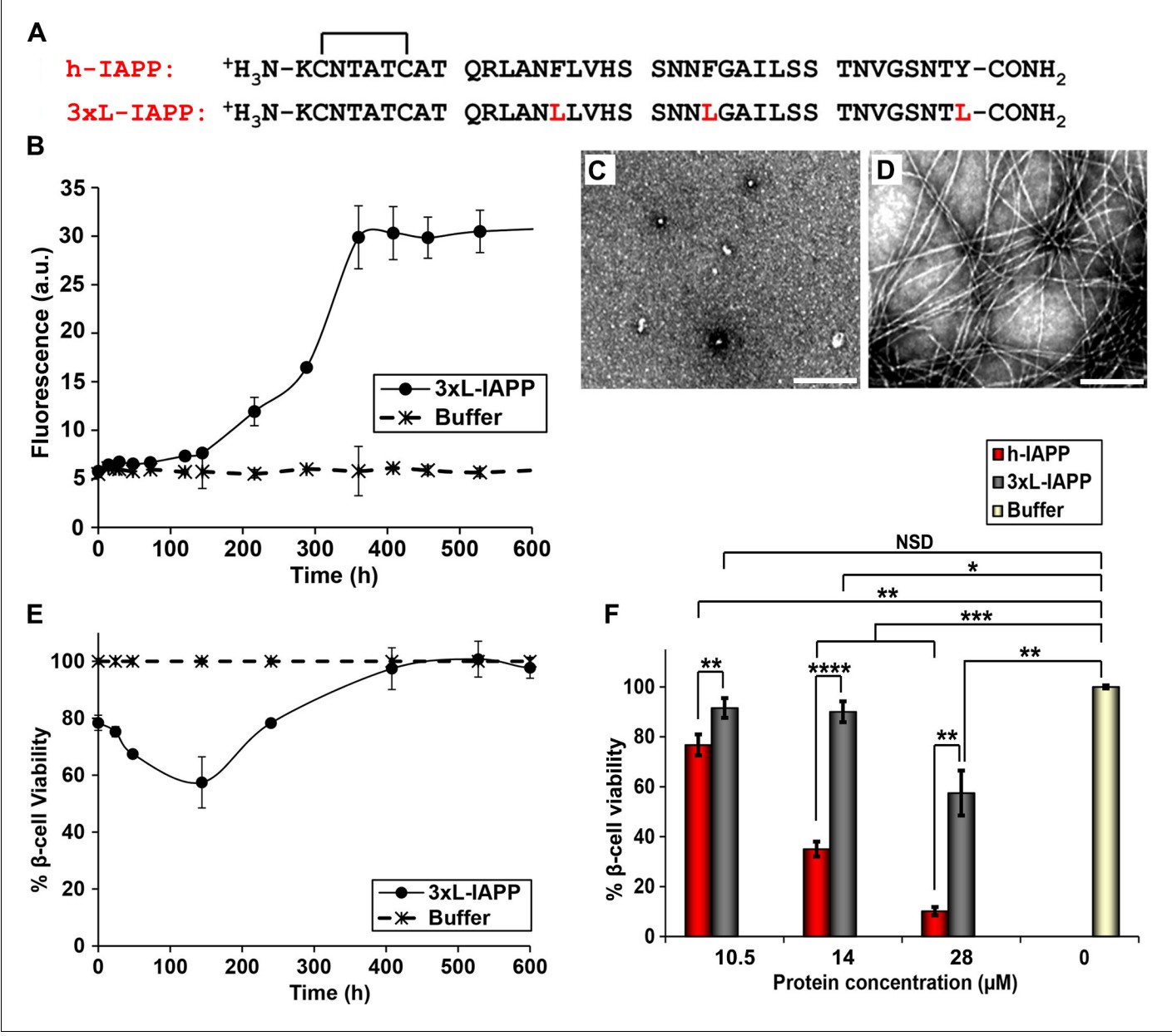

**Figure 9.** Aromatic-aromatic and aromatic-hydrophobic interactions are not required for toxicity. (**A**) Primary sequences of h-IAPP and 3xL-IAPP. Amino acid positions differing from h-IAPP are indicated in red. (**B**) Thioflavin-T monitored kinetics of amyloid formation by 3xL-IAPP (•) and buffer control (∗). (**C**) TEM image of spherical, toxic, mid-lag phase intermediates produced during amyloid formation by 3xL-IAPP. (**D**) TEM image of non-toxic amyloid fibrils produced by 3xL-IAPP. (**E**) Time-resolved Alamar Blue reduction assays of β-cells treated with 3xL-IAPP (•) or buffer (∗) at different time points during the course of amyloid formation. (**F**) Alamar Blue reduction assays measuring β-cell viability in response to increasing doses of h-IAPP or 3xL-IAPP with respect to buffer treated cells: h-IAPP (red), 3xL-IAPP (dark grey) and buffer (gold). Concurrent Alamar Blue reduction assays, thioflavin-T binding assays and TEM studies were carried out using aliquots from the same 40 µM peptide solutions. The peptide concentration in samples assessed in panels **B**–**D** was 40 µM. The final peptide concentration in samples assessed in panel **E** after dilution of the 40 µM peptide solutions into β-cell assays was 28 µM. The final peptide concentrations in dose-response experiments in panel **F**, after dilution of peptide solutions into β-cell assays was 10.5, 14 and 28 µM. Data represent mean ± SD of three to six replicate wells per condition and three replicate experiments per group. Some of the error bars in panels **B** and **E** are the same size or smaller than the symbols in the graphs (Scale bars: 200 nm; *p<0.05, **p<0.01, ***p<0.001, ****p<0.0001).

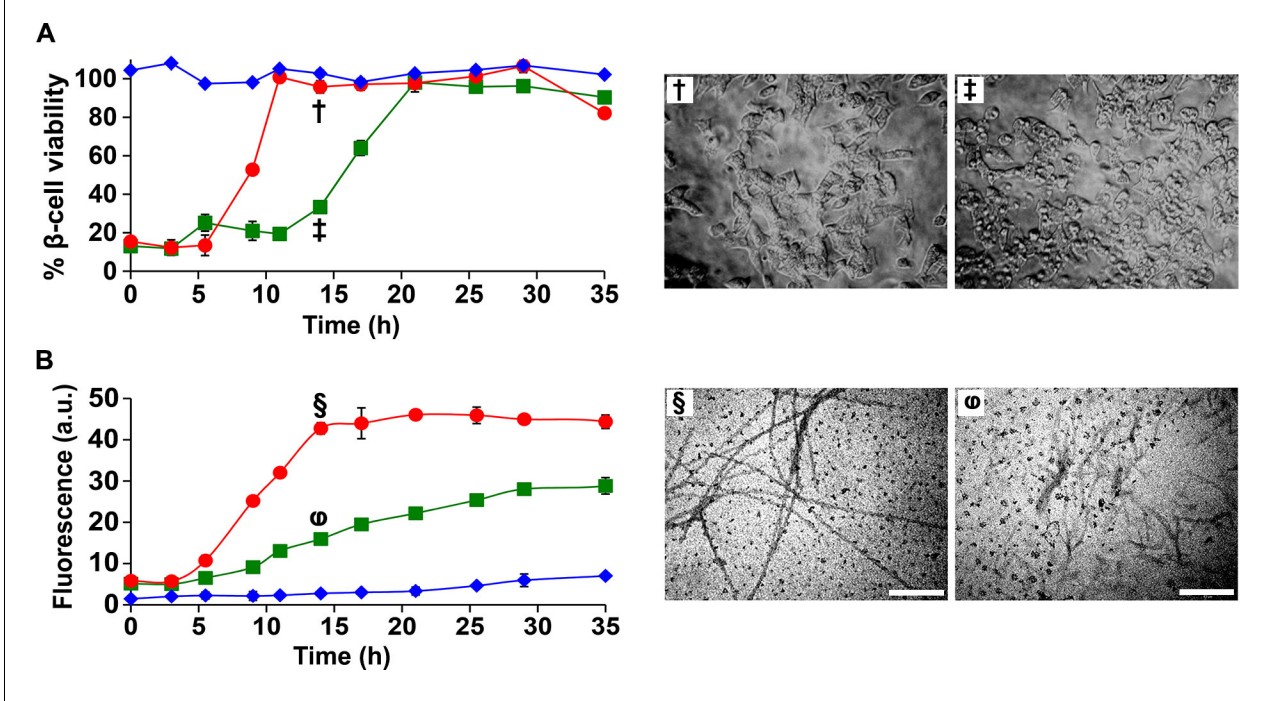

**Figure 10.** I26P-IAPP inhibits h-IAPP amyloid formation, but prolongs cytotoxicity. (**A**) Time-resolved Alamar Blue reduction assays of β-cells treated with: I26P-IAPP (♦), h-IAPP (•) and 1:1 I26P-IAPP/h-IAPP (■). Light microscopy: (†) Viable β-cells treated with h-IAPP amyloid fibrils formed after 14 h- of incubation; (‡) shrunken apoptotic β-cells after treatment with lag phase intermediates of 1:1 I26P-IAPP/h-IAPP produced after 14 h of incubation. (**B**) Thioflavin-T monitored kinetics of amyloid formation by I26P-IAPP (♦), h-IAPP (•) and 1:1 I26P-IAPP/h-IAPP (■). The same color coding is used in panels A and B. TEM images of aliquots of: (§) h-IAPP and (φ) 1:1 I26P-IAPP/h-IAPP obtained from the same samples monitored in panel B and applied to β-cells in panel A (Scale bars: 200 nm). Concurrent Alamar Blue reduction assays, thioflavin-T binding assays, light microscopy and TEM studies used aliquots from the same peptide solutions containing 40 μM (single peptide samples) or 80 μM peptide (1:1 mixture). The final peptide concentrations after dilution into β-cell assays were 28 μM and 56 μM, respectively. Data represent mean ± SD of three to six replicate wells per condition and three replicate experiments per group. Some of the error bars in panels A and B are the same size or smaller than the symbols in the graphs.

### Inhibitors of amyloid formation may paradoxically stabilize toxic conformations and prolong cytotoxicity

The observation that h-IAPP toxicity is directly induced by pre-fibrillar lag phase species highlights these species as a key drug target for inhibitor design, and predicts that an inhibitor that slows the onset of amyloid formation, but does not prevent it could actually prolong cytotoxicity by prolonging the lifetime of the toxic intermediates. This is particularly important since many in vitro screens of amyloid inhibitors rely on kinetic assays of amyloid formation. We tested this hypothesis using the I26P-IAPP inhibitor (*Abedini et al., 2007*). Time-resolved kinetic studies of cytotoxicity and amyloid formation show that a 1:1 addition of I26P-IAPP lengthens both the lag phase and the growth phase, and increases the duration of toxicity proportionally (*Figure 10*). The value of T50 (the time required to reach 50% of the total signal change in a thioflavin-T experiment) is increased by a factor of two, as is the length of the lag phase, defined here as the time required to reach 10% of the total change in thioflavin-T signal. We conclude that an effective inhibitor of amyloid formation can be deleterious to cells if it does not prevent the formation or the accumulation of toxic lag phase intermediates, but traps them instead in their toxic conformation.

## Discussion

In the present work, we use a combination of biophysical, biochemical and cell biological techniques to define the basis of h-IAPP induced islet amyloidosis toxicity. Toxic h-IAPP species are found to be partially structured, globally flexible, soluble, low order oligomers with solvated aromatic side chains

that populate the lag phase of amyloid formation. The data do not preclude the ordering of short segments of the chain, nor do they rule out regions with intermolecular hydrogen bonds or short segments of intermolecular β-sheets. The ensemble of toxic h-IAPP intermediates are susceptible to proteolysis and do not require π-π interactions or aromatic-hydrophobic contacts to form, however removal of the aromatic residues does reduce toxicity. These toxic species of h-IAPP induce oxidative stress and pro-inflammatory cellular processes leading to β-cell apoptosis. Studies with I26P-IAPP, TM-IAPP and r-IAPP demonstrate that not all IAPP oligomers are toxic, decoupling general oligomerization from toxicity; and suggest that the conformational properties of oligomers and/or their stability, rather than their size, are important determinants of toxicity. Along these lines, IM-MS studies indicated that low order r-IAPP and h-IAPP oligomers have different gas phase conformations and exhibit different gas phase stabilities, and suggest that certain inhibitors of toxicity target subspecies of oligomers (*Young et al., 2014*; *Dupuis et al., 2009*). Toxicity has been linked to the surface hydrophobicity of oligomers formed by other amyloidogenic proteins (*Mannini et al., 2014*), but that does not appear to rationalize the relative toxicity of the oligomers examined here. The mutations do reduce the hydrophobicity of the chain, but the effects are modest, particularly for the I26P-IAPP point mutant, and h-IAPP oligomers do not bind the dyes ANS, bis-ANS or Nile Red (*Figure 5—figure supplements 6* and *7*, *Figure 7*, *Figure 7—figure supplement 1*). It is not currently possible to pinpoint the structural features that distinguish toxic oligomers from those that are non-toxic, but all of the non-toxic variants examined here contain proline substitutions within a region of h-IAPP that has been postulated to form transient, parallel, in register β-sheet structure during amyloid formation in solution. The ability of proline to disrupt secondary structure in this region may well be an important feature in reducing toxicity.

Toxic h-IAPP oligomers share some features with toxic oligomers reported to be produced by other amyloidogenic proteins, but also have distinct molecular properties. Like other toxic oligomers, particularly those that are formed by intrinsically disordered proteins, they are soluble, pre-fibrillar in character, contain partial secondary structure, and bind to molecules such as EGCG. However, they do not bind ANS, bis-ANS or Nile Red; and the overall β-sheet and α-helical content measured for the ensemble of toxic h-IAPP oligomers in solution is much less than that described for a range of other amyloidogenic proteins. Isoforms of the Aβ peptide of AD and a fragment derived from αB crystallin have been shown to contain extensive regions of β-sheet structure, while oligomers formed from several other proteins are reported to be rich in α-helical structure (*Chiti and Dobson, 2006*; *Bolognesi et al., 2010*; *Chimon et al., 2007*; *Glabe, 2008*; *Kim et al., 2009*; *Laganowsky et al., 2012*; *Mannini et al., 2014*; *Lendel et al., 2014*; *Sandberg et al., 2010*; *Stroud et al., 2012*; *Chen et al., 2015*; *Sarkar et al., 2014*). Thus, these data reveal, for the first time, that the toxic h-IAPP intermediates differ from toxic oligomers described in other amyloidoses (*Campioni et al., 2010*; *Bolognesi et al., 2010*; *Chimon et al., 2007*; *Glabe, 2008*; *Kim et al., 2009*; *Laganowsky et al., 2012*; *Mannini et al., 2014*; *Lendel et al., 2014*; *Lorenzen et al., 2014*; *Sarkar et al., 2014*).

The differences between the conformational properties of toxic h-IAPP oligomers defined in the present work and those recently identified for Aβ are particularly interesting, given that the two polypeptides share important features and given the growing evidence that links T2D and AD (*Yang and Song, 2013*; *Oskarsson et al., 2015*; *Ninomiya, 2014*). Pre-fibrillar forms of h-IAPP and Aβ interact in vitro and a positive association has recently been demonstrated in plasma (*Miklossy et al., 2010*; *Qiu et al., 2014*). Furthermore, Aβ can seed amyloid formation by h-IAPP in vitro and Aβ has been reported to form pancreatic deposits in T2D, while h-IAPP has been reported in brain plaques in AD (*O'Nuallain et al., 2004*; *Oskarsson et al., 2015*; *Miklossy et al., 2010*). The data presented here demonstrate that toxic species formed by different proteins can be distinct in their biophysical/biochemical properties, even when the two sequences share common features, helping to rationalize why some inhibitors of h-IAPP amyloid formation do not inhibit Aβ amyloid formation and vice versa (*Rochet, 2007*; *Wang and Raleigh, 2014*). This is important since it indicates that therapeutic strategies for amyloidosis diseases need to be tailored to the specific molecular properties of pathological amyloidogenic species unique to each disease.

We demonstrate that some inhibitors of amyloid formation can adversely prolong toxicity depending on their targets and their modes of action. Inhibitors that stabilize the ensemble of toxic lag phase intermediates can trap them and exacerbate pathological cellular cascades. This observation provides additional evidence that toxicity appears to be conformation dependent, and has

implications for rational drug design for the treatment of amyloidosis diseases. The findings also emphasize that caution must be taken when in vitro biophysical assays are used, such as thioflavin-T binding, to develop leads for anti-amyloid agents, since drugs that slow the onset of amyloid formation and cause the buildup of toxic pre-fibrillar intermediates can give the same spectroscopic signatures as compounds that slow amyloid formation, but decrease the steady state population of toxic species. Conversely, compounds that accelerate amyloid formation could reduce toxicity by reducing the transient population of toxic oligomers (*Bieschke et al., 2012*).

This work illustrates the power of combining time-resolved kinetic studies with physio-chemical, biochemical and biological measurements, and highlights that while amyloid formation by different proteins may share many common features, toxic species produced by different proteins can have different properties. In the case of pancreatic islet amyloidosis, we conclude that flexible, low order, toxic h-IAPP oligomers with modest overall β-sheet and α-helical content, which form before amyloid fibrils, are primary targets for therapeutic interventions. Hence, molecules that decrease the population of toxic amyloidogenic species by preventing their formation, reducing their lifetime, or sequestering them to prevent their interactions with cells may serve as therapeutic agents in disorders characterized by pancreatic β-cell dysfunction, and more broadly to a wide range of other protein misfolding diseases (*Westermark et al., 2011*; *Campioni et al., 2010*; *Johnson et al., 2012*; *Blancas-Mejía and Ramirez-Alvarado, 2013*).

## Materials and methods

### Protein preparation

h-IAPP, r-IAPP and IAPP analogs were prepared using Fmoc chemistry and pseudoproline derivatives as previously described (*Abedini and Raleigh, 2005*; *Abedini et al., 2006*; *Marek et al., 2010b*) or purchased from the KECK Foundation at Yale University. Peptides were cleaved from the resin using standard TFA methods. The peptide disulfide bond was formed via DMSO-based oxidation and peptide purification was achieved by reverse phase HPLC using a C18 preparatory column (*Abedini et al., 2006*; *Marek et al., 2010b*). HCl, rather than TFA, was used as the ion pairing agent since residual TFA can affect IAPP amyloid formation kinetics and can also interfere with 2D IR studies. Samples were analyzed by MALDI-TOF Mass Spectrometry (Brucker) or by Electrospray Mass Spectrometry using a Micromass Platform LCZ single quadrupole instrument to confirm their identity.

### Cell culture

Rat INS-1 β-cells (832/13) were generously provided by Professor Newgard (Duke University). β-cells were grown in RPMI 1640 supplemented with 10% fetal bovine serum (FBS), 11 mM glucose, 10 mM Hepes, 2 mM L-glutamine, 1 mM sodium pyruvate, 50 μM β-mercaptoethanol, 100 U/ml penicillin, and 100 U/ml streptomycin.

### Islet isolation and culture

Pancreatic islets were isolated from anesthetized 12–18 week-old male C57BL/6 mice (Jackson Laboratories) according to institutional guidelines by ductal collagenase injection, oscillating digestion, and filtration through a 70 μm filter. Hand-picked murine islets were assessed by light microscopy and immunofluorescence to insure intact mantels, insulin-positivity and absence of inflammation prior to experiments. Islets were seeded at 25–30 islets per well in RPMI 1640 supplemented with 10% fetal bovine serum (FBS), 11 mM glucose, 10 mM Hepes, 2 mM L-glutamine, 1 mM sodium pyruvate, 100 U/ml penicillin and 100 U/ml streptomycin.

### Immunofluorescence (IHC-IF)

Formalin-fixed, paraffin-embedded pancreas specimens were cut into sections 4 μm thick, and six sections, 30 μm apart, were labeled for each marker. All sections were co-stained with anti-insulin antibody (1:300, Dako) to visualize β-cells and anti-F4/80 antibody (1:75, Cedarlane) to detect macrophages and thus assess inflammation. Staining of tissue was carried out by blocking pancreatic sections in PBS containing 2.0% normal goat serum (Vector Laboratories) and incubating with primary antibody diluted in PBS/1% BSA, followed by incubation with secondary antibody diluted in

PBS for 1 h. Secondary antibodies for immunolabeling of insulin (1:100, Alexa Fluor 594-conjugated goat anti-guinea pig immunoglobulins) and F4/80 (1:100, Alexa Fluor 488-conjugated goat anti-rat immunoglobulins) were all purchased from Invitrogen. Sections were counterstained with Dapi anti-fade mounting media (2 µg/mL; Invitrogen) to identify nuclei. Images were taken using a Leica fluorescent microscope.

## Amyloid formation and aggregation assays

Synthetic amyloidogenic and non-amyloidogenic IAPP peptides were dissolved in 1,1,1,3,3,3-Hexafluoro-2-propanol (HFIP) for 5–12 h, aliquoted and lyophilized to a dry powder. Amyloid formation was initiated by dissolving dry HFIP-treated peptides in 20 mM Tris HCl buffer (time-zero) at pH 7.4. Samples were incubated at 25°C, unless indicated otherwise. Aliquots were removed from the peptide solutions at designated time points over the course of amyloid formation or aggregation as schematically depicted in *Figure 2A*. The kinetic species populated within each of the three phases of amyloid formation were concurrently characterized by biophysical methods and biological assays assessing changes in metabolism, oxidative stress, inflammation and apoptosis using INS-1 β-cells and murine pancreatic islets.

## Toxicity assays

Rat INS-1 β-cells were seeded at a density of 30,000 cells per well in 96-well plates 12–16 h prior to start of experiments. Hand purified islets were seeded at a density of 25–30 islets per well in 96-well plates. Amyloid formation and aggregation assays were initiated by dissolving IAPP peptides (15 µM to 80 µM stock solutions for dose-response experiments) in 20 mM Tris HCl (pH 7.4), unless stated otherwise. Peptide solutions were incubated at 25°C, or as indicated. Aliquots were removed from amyloid formation and aggregation assays at different time points and applied exogenously to rat INS-1 β-cells (5 h incubation) or murine pancreatic islets (10 h incubation). β-cells and islets were photographed by light microscopy immediately before and after toxicity experiments to assess changes in morphology. The final range of peptide concentrations examined in β-cell and islet toxicity assays after diluting IAPP peptide solutions into cell or islet culture was 10.5 µM to 56 µM in dose-response studies, unless indicated otherwise. Cell viability was measured by Alamar Blue reduction assays, which detect changes in metabolic function, and by morphological changes detected by light microscopy. Alamar Blue was diluted ten-fold in culture medium and incubated on β-cells or islets for 5 h at 37°C. Fluorescence (530 nm excitation and 590 nm emission) was measured with a Beckman Coulter DTX880 plate reader. Values were calculated relative to control β-cells or islets treated with buffer alone. Toxicity was defined as <80% viability. Light microscopy images were captured using an Olympus BX-61 light microscope.

## Oxidative stress assays

INS-1 β-cells were treated with h-IAPP lag phase intermediates (14 µM or 28 µM final concentration on cells) or Tris HCl buffer control solutions for 1 h. A shorter solution incubation time on cells was employed in oxidative stress experiments (1 h) than employed in standard toxicity experiments (5 h), since the production and detection of transient reactive oxygen species (ROS) occurs prior to detection of loss in cell viability. Superoxide production was measured with dihydroethidium (DHE), a cell-permeable dye that fluoresces upon binding of intracellular superoxide anions. DHE was added to cells (40 µM final concentration) and incubated on cells during the last 30 min of incubation with h-IAPP or control solutions. Fluorescence was subsequently measured (518 nm excitation and 605 nm emission). Data were normalized to cell number detected by the Calcein AM live cell assay. NOX1 protein expression was also assessed in cell lysates produced from β-cells treated with either h-IAPP or buffer, via western blot using anti-NOX1 antibody (1:500, abcam). Western blot data was normalized to GAPDH levels detected by anti-GAPDH antibody (1:1000, abcam).

## Calcein AM live cell assay

Calcein AM is cleaved to a fluorescent byproduct after interaction with viable intracellular esterases and thus fluorescence approximates the proportion of viable cells per well. Following incubation with h-IAPP or control solutions, media was removed and cells were incubated with 5 µM Calcein

AM in phosphate buffered saline (PBS) for 15 min prior to reading fluorescence (485 nm excitation and 535 nm emission).

## Apoptosis assays

Rat INS-1 β-cells were seeded at a density of 500,000 cells per well in 6-well plates 24 h prior to start of experiments. Aliquots were removed from amyloid formation assays or aggregation assays (20 μM peptide) at different time points and transferred to cultured β-cells. Final peptide concentration after dilution into cellular assays was 14 μM. β-cells were lysed and protein extracts were assessed by caspase-3 colorimetric assay (R&D Systems). Recombinant caspase-3 enzyme (R&D Systems) was used as a positive control.

## RNA isolation and quantitative real time PCR (qPCR)

Total cellular RNA was isolated from h-IAPP treated β-cells using the RNeasy Plus Mini Kit (Qiagen). The quality of RNA was determined by measurement of 260:280 ratio. One μg of RNA was reverse-transcribed to cDNA using MultiScribe reverse transcriptase (Applied Biosystems). Real-time quantitative PCR was performed using the TaqMan method (50°C for 2 min, 95°C for 10 min, and 40 cycles of 95°C for 15 s and 60°C for 1 min) with premade *Ccl2* and *Il1b* primers (Life Technologies). The relative mRNA contents were normalized according to the expression of 18S rRNA using the ΔΔCt method. qPCR was carried out using an Applied Biosystems 7500 Real Time PCR machine.

## Thioflavin-T binding assays

Aliquots (100 μL) were removed from amyloid formation and aggregation assays at different time points, and added to 96-well plates containing 8 μL of a 1 mM thioflavin-T solution. Fluorescence was measured using a Beckman Coulter DTX880 plate reader (445 nm excitation and 485 nm emission). Final solution conditions contained 16 mM Tris HCl and 74 μM thioflavin-T (pH 7.4).

## Transmission electron microscopy (TEM)

Aliquots (4 μL) were removed from amyloid formation or aggregation assays at different time points and placed on a carbon-coated 200-mesh copper grid and negatively stained with saturated uranyl acetate. The samples were imaged with a Philips CM12 or a FEI BioTwinG2 transmission electron microscope.

## Far UV CD

Far UV CD was performed using an Applied Photophysics circular dichroism spectrophotometer. Aliquots (300 μL) were removed from amyloid formation or aggregation assays at different time points and transferred to a 0.1 cm quartz cuvette a few minutes prior to data collection. Spectra were recorded over a range of 190 to 260 nm, at 1 nm intervals with an averaging time of 3 s. CD spectra represent the average of five repeats. Background spectra were subtracted from collected data. Samples contained 20 mM Tris HCl (pH 7.4).

## 2D IR

2D IR spectra of 40 μM h-IAPP solutions were recorded using 60 femtosecond (full width at half maximum) mid-IR pulses generated by a Ti:sapphire femtosecond laser system and a mid-IR pulse shaper, as previously described (*Middleton et al., 2010*; *Shim et al., 2006a*; *2006b*).

## ANS and *p*-CyanoPhe fluorescence measurements

*p*-CyanoPhe (240 nm excitation, 296 nm emission) and ANS fluorescence (370 nm excitation and 460 nm emission) were measured using a Photon Technology International instrument. ANS binding studies were conducted by adding aliquots from amyloid formation assays (20 μM peptide solutions) at different time points to a cuvette containing ANS. Final sample conditions contained 16 mM Tris HCl and 10 μM ANS (pH 7.4).

## Photochemical-Induced Cross-Linking

Samples were cross-linked using Tris(bipyridyl)Ru(II), in the presence of ammonium persulfate. IAPP peptides were incubated for indicated times in 20 mM Tris HCl (pH 7.4) at 25°C. Aliquots were

removed at indicated time points, centrifuged at 20,000 $g$ for 20 min and added to the cross-linking solution. Samples contained a final concentration of 20 µM peptide, 70 µM Tris(bipyridyl)Ru(II) and 1.4 mM ammonium persulfate. Samples were illuminated with a 150 W incandescent bulb for 5 s, unless otherwise noted. The reaction was quenched by the addition of β-mercaptoethanol. The products were separated by SDS polyacrylamide gel electrophoresis using a 10–20% Tris-tricine gradient gel. Oligomer bands were visualized by silver staining (SilverXpress, Invitrogen). Quantitative analysis was carried out using GelAnalyzer software version 2010a. The relative intensity of each band was calculated by first correcting the baseline, then integrating the area under each peak.

### Solubility assays

Samples of toxic lag phase intermediates or amyloid fibrils were ultracentrifuged for 20 min (20,000 $g$) and the protein in the soluble fraction (the supernatant) was measured by UV absorbance (215 nm) before and after ultracentrifugation using a DU 730 Life Science UV/Vis spectrophotometer (Beckman Coulter). The soluble phase species were further characterized, before and after centrifugation by CD, TEM, thioflavin-T binding assays and toxicity assays. r-IAPP, which does not form amyloid, was used as a control.

### Proteolytic digestion assays

h-IAPP was incubated in 20 mM Tris HCl buffer (pH 7.4) at 25°C. Aliquots were removed at various times and incubated with Proteinase K for 5 or 40 min at 37°C. The solutions were desalted, mixed with an equal volume of $\alpha$-cyano-4-hydroxycinnamic acid matrix and spotted on a MALDI-TOF MS plate for analysis.

### Statistical analysis

Data represent mean ± SD or mean ± SEM of three to six technical replicates per condition and a minimum of three to ten biological replicate experiments per group. Differences between two groups were evaluated using the Student t-test of two samples assuming unequal variances. A p-value of $\leq 0.05$ was considered significant.

## Acknowledgements

We thank Professor Christopher Newgard for generously providing us with rat INS-1 β-cells, Dr. Peter Marek for help with sample preparation, Dr. Jacqueline Lonier for experimental assistance, and Ms. Latoya Woods for help with manuscript preparation.

## Additional information

### Funding

| Funder | Grant reference number | Author |
| --- | --- | --- |
| National Institutes of Health | 1F32DK089734-02 | Andisheh Abedini |
| National Institutes of Health | HL60901 | Andisheh Abedini<br>Jinghua Zhang<br>Daniel J Sartori<br>Ann Marie Schmidt |
| Canadian Institutes of Health Research | MOP-14682 | Annette Plesner<br>C Bruce Verchere |
| Canadian Diabetes Association | OG-3-11-3413-CV | Annette Plesner<br>C Bruce Verchere |
| National Institutes of Health | GM078114 | Ping Cao<br>Zachary Ridgway<br>Ling-Hsien Tu<br>Fanling Meng<br>Hui Wang<br>Amy G Wong<br>Daniel P Raleigh |
| National Institutes of Health | 5T32GM09271405 | Zachary Ridgway |

| National Institutes of Health | DK79895 | Chris T Middleton<br>Martin T Zanni |
| National Institutes of Health | 2T35 DK007421 | Brian Chao |

The funders had no role in study design, data collection and interpretation, or the decision to submit the work for publication.

## Author contributions

AA, Designed the research, Conducted experiments, Analyzed data, Wrote the manuscript; AP, PC, ZR, JZ, L-HT, CTM, BC, DJS, HW, AGW, Conducted experiments, Analyzed data; FM, Prepared critical reagents; MTZ, CBV, Designed experiments, Analyzed data; DPR, AMS, Designed and directed research, Analyzed data, Wrote the manuscript

## Author ORCIDs

Ann Marie Schmidt, http://orcid.org/0000-0001-8902-070X

## Ethics

Animal experimentation: All procedures were approved by the Institutional Animal Care and Use Committee of New York University Langone Medical Center (NYULMC) and conform to the Guide for the Care and Use of Laboratory Animals published by the US National Institutes of Health (NIH) (8th Edition, 2011, ISBN 10: 0-309-15400-6). The Animal Care and Use Program at NYULMC is in full compliance with NIH policy (NYULMC Compliance Number is A3435-01).

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
