## [Decision Letter]

Thank you for submitting your work entitled "Time-resolved studies define the nature of toxic IAPP intermediates, providing insight for anti-amyloidosis therapeutics" for consideration by *eLife*. Your article has been reviewed by two peer reviewers, and the evaluation has been overseen by Reviewing Editor Jeffery W. Kelly and Richard Aldrich as the Senior Editor. One of the two reviewers has agreed to reveal his identity: William F. De Grado (peer reviewer).

The reviewers have discussed the reviews with one another and the Reviewing Editor has drafted this decision to help you prepare a revised submission.

Summary:

In this manuscript, the authors describe a kinetic study of IAPP amyloid formation and toxicity. In particular, they use several biochemical and spectroscopic tools to characterize the kinetic progression of aggregation, and they correlate these results with the time-dependence of toxicity.

Essential revisions:

Please carefully consider these points in the revision in the context of the reviews that can be found below:

1) How can the authors confirm that lower populations of larger oligomers (low intensity smear on gel) are not the toxic species?

2) What structure is in which oligomeric form?

3) At the very least recognize that the structures applied to cells may be remodeled before toxicity is observed.

4) That dilution does not alter the kinetics of aggregation and needs more clarification – what does this imply for the mechanism since it is not a unimolecular process?

5) Address the first series of questions from reviewer 2 focused on: Some discussion of how far this is from a natural system is warranted, particularly focusing on the effects of: 1) steady state production in vivo versus the artificial initiation in the present work; 2) are the high concentrations created in this paper relevant to what is seen in vivo; 3) would these be toxic if tested in isolated tissue or an animal model? Clearly, it is not possible to address all these issues experimentally, but the authors should clearly state the extent of the limitations of their system.

6) Address experimental questions 1 and 2 from Reviewer 2 regarding the CD and FT-IR structural conclusions central to the paper.

7) Address experimental question 3 from Reviewer 2 regarding the cross-linking controls that are critical to address in the revision.

8) The authors argue that the oligomers are held together by highly flexible interactions that would be expected to equilibrate rapidly. However, they show experimentally that there is essentially no change in the cross-linking pattern when oligomers are diluted and allowed to equilibrate. One would expect that this would lead to extensive dissociation of the oligomers (note that the stability of an n-mer depends on the nth value of the monomer concentration). Thus, one might expect rapid dissociation from even a modest dilution. It is surprising that the authors fail to discuss this apparent paradox.

Reviewer #1:

In this manuscript, the authors describe a kinetic study of IAPP amyloid formation and toxicity. In particular, they use several biochemical and spectroscopic tools to characterize the kinetic progression of aggregation, and they correlate these results with the time-dependence of toxicity. The study results in the conclusion that fibrillar species are not toxic, but pre-fibrillar ones are. Furthermore, these species are deduced to be small-size oligomers, with low levels of β-sheet structure and exposed hydrophobic surfaces. Interestingly, compounds that inhibit amyloid formation increase toxicity.

Overall, this work is on an important structural question related to IAPP toxicity in T2D. A number of spectroscopic and biochemical tools together provide some interesting information about the structural, oligomeric and toxicity states of the system as it progresses from monomer to amyloid. The effects of the anti-amyloid compounds are also important. The paper is systematically laid out. These strengths would make the work of interest to the readers of *eLife*. However, the work has some issues that need to be addressed. Therefore, I feel the paper is not currently at a level meriting publication in *eLife*.

Most importantly, the structural information provided by the work is somewhat limited. Little specific information about what structure is in which oligomeric species is obtained. While cyanoPhe and ANS experiments are good, they again do not provide specific information. The 2D IR studies are most informative, but it is hard to correlate them with specific species. Furthermore, the authors should provide more information about the crosslinking studies (below).

With the crosslinking studies, the control with HP35 is not quite convincing. Additional transient non-specific interactions between monomers for IAPP might also give rise to some the observed crosslinking differences. The non-toxic mutant studies may also provide some information on this aspect of the study. Authors please discuss this aspect and what they characterize as oligomeric species in more detail.

How can the authors confirm that lower populations of larger oligomers (low intensity smear on gel) are not the toxic species?

Related to the above point, toxic species may involve rearrangements upon interaction with cell environments and may not be the starting species that was characterized biophysically. However, this is a general problem with many such correlations in the field and the current results are still interesting.

The dilution experiment in Figure 2—figure supplement 1 seems to indicate that some of the smaller species are stable over time (it should be noted that the larger species smear is reduced by dilution) – error bars are missing for the orange bars.

That dilution does not alter the kinetics of aggregation and needs more clarification – what does this imply for the mechanism since it is not a unimolecular process?

Reviewer #2:

This is an interesting paper describing biophysical characterization of h-IAPP oligomers. This reviewer has many questions that should be addressed prior to publication.

1) Toxicity is shown in vitro to β-cells at a concentration of 14 microM. Also, the peptide is dissolved at slightly higher concentration, and then the kinetics of the process is measured. This would appear to be an artificial system, and it is not clear how much of an extrapolation is needed to approximate the in vivo situation. What concentration of h-IAPP is encountered in the relevant tissue? It would appear that at steady state in an animal the concentration of oligomers would be extremely low because, as the authors point out, they convert quickly to fibrils that are not toxic? Are the oligomers toxic to β-cells in tissue? Some discussion of how far this is from a natural system is warranted, particularly focusing on the effects of: 1) steady state production in vivo versus the artificial initiation in the present work; 2) are the high concentrations created in this paper relevant to what is seen in vivo; 3) would these be toxic if tested in isolated tissue or an animal model? Clearly, it is not possible to address all these issues experimentally, but the authors should clearly state the extent of the limitations of their system.

Experimental Questions:

1) The authors use CD to characterize the oligomeric species. Large light-scattering species are removed by centrifugation to enable the analysis. They state that the band appears similar to the α-helix, but the data are recorded in raw mdeg rather than mean residue ellipticity. If the sample is indeed helical, then one would expect a very strong mean residue ellipticity. This conversion requires knowledge of the concentration; it is possible to determine it to reasonable precision from the amide absorption (if it is possible to measure a CD spectrum it is also possible to measure an absorption spectrum, see Scopes Anal. Biochem. 59, 277, 1974 for the method). The authors should comment on whether the CD spectrum indeed has the correct intensity for an α-helix. If it is some small fraction of the expected value then it would follow that the similarity to an α-helix is not meaningful, because partially helical spectra would have a different shape.

2) "Two dimensional infrared (2DIR) studies, described below, indicate that the level of β-structure is modest." Clearly, the amount of very regular β-structure (that would give rise to strong excitonic coupling) is very low from this experiment. However, it is not clear that there is sufficient concentration of protein to rule out less repetitive and partially hydrated β-structure as would be seen in globular water-soluble proteins. The authors have extensive experience in this area, so I don't doubt their conclusions. However, they should include comparisons to other literature studies from their or other groups to support this statement. I would suggest being careful with language, if the authors intend to simply rule out the possibility of the type of β-structure seen in amyloids vs. the less regular (but not necessarily disordered) β-structure seen in globular proteins.

3) The use of cross-linking to probe oligomer stoichiometry is lacking in appropriate controls. Firstly, according to the experimental procedures the cross-linking is performed on the peptide solution without centrifugation. This means that the great majority of what is submitted to crosslinking is in the form of a large insoluble aggregate for h-IAPP but not for r-IAPP (earlier in the paper the authors state: "Samples of toxic h-IAPP intermediates and amyloid fibrils were pelleted at 20,000 g for 20 min and the soluble peptide remaining in the supernatant was measured…. At least 88% of h-IAPP is pelleted in the sample of fibrils, even at these low g-forces, while 94% of the peptide in the sample of toxic lag phase intermediates remains in the supernatant"). It is unclear why the samples are not centrifuged.

4) The natural control for this experiment is to measure the extent of crosslinking near t=0, at the maximum toxicity time for the experimental conditions, and at long time after conversion to fibrils. The data at long and short time are entirely missing, and the paper should not be published without them, as the experiments would otherwise be misleading.

5) Moreover, the authors need to factor in the cross-linking efficiency. They should provide simulations of the distribution of species expected for an infinite chain of interacting monomers with a probability of crosslinking ranging from 0.7 to 0.95 so the reader can see the sensitivity of their conclusions to the cross-linking efficiency.

6) Finally, the authors argue that the oligomers are held together by highly flexible interactions that would be expected to equilibrate rapidly. However, they show experimentally that there is essentially no change in the cross-linking pattern when oligomers are diluted and allowed to equilibrate. One would expect that this would lead to extensive dissociation of the oligomers (note that the stability of an n-mer depends on the nth value of the monomer concentration). Thus, one might expect rapid dissociation from even a modest dilution. It is surprising that the authors fail to discuss this apparent paradox.

In conclusion, I would not pretend to be an expert in h-IAPP toxicity. Nevertheless, I was left with many questions, that it seems could easily be addressed in a revision.

Reviewer #2 (Additional data files and statistical comments):

I indicated in the review that there are inadequate controls to allow interpretation of the cross-linking data.

---

## [Author Response]

Summary of Additional Studies Completed for Revision:

I) We conducted additional biological experiments to demonstrate that h-IAPP lag phase intermediates produced in vitro are toxic to murine pancreatic islets. We believe that these studies are important since they provide direct evidence that the oligomeric intermediates are toxic to cells in tissue. The new results are fully consistent with the studies of cultured cells found in the original submission. New panels have been added to Figure 2 to illustrate this data.

II) Although not requested, we carried out experiments using additional conformationally sensitive dyes. Experiments with bis-ANS and Nile Red fully support the conclusions of the ANS studies found in the original submission. The new data is presented as a new figure supplement. These studies, as well as some of the other additional experiments, were carried out by Ms. Wong. They represent a large amount of work thus we have added her to the author list.

III) We repeated the cross-linking studies presented in the main figures and several presented in the figure supplements of our original submission using newly synthesized batches of peptide. These studies demonstrate that the data is very reproducible and also argue that the “smear” seen in some of the original gels likely arose, in part, because of technical staining issues.

IV) We conducted new cross-linking studies to test if the observed distributions are robust to the time used in the cross-linking studies. The new control experiments show that the detection of monomers through hexamers is not a consequence of the choice of irradiation time. The data also further argues that the low intensity “smear” seen on some of the original gels did not represent lower populations of larger oligomers that are the toxic species. The new data is presented as a new figure supplement.

V) We conducted additional cross-linking studies to examine the relative distribution of oligomers in the saturation phase of amyloid formation (i.e. time points at which amyloid formation appears completed as judged by thioflavin-T assays and TEM). The new experiments show that the saturation phase is populated by high molecular weight fibrils that are too large to migrate down the denaturing gel in SDS-PAGE, as well as monomers and small amounts of dimers. The new data is shown as a new figure supplement.

VI) New cross-linking studies were conducted to assess the relative distribution of oligomers at “time-zero”. Time-zero corresponds to several minutes. The relative distribution of oligomers is similar to the distribution observed during lag phase, in agreement with independent mass spectroscopy studies. The new data is presented as a new figure supplement.

VII) We repeated the CD studies presented in the main figures of the original submission. The data is highly reproducible. We recalculated the original CD data and present them as mean residue ellipticity. These new plots replace the old plots in two of the figures in the revised manuscript.

VIII) We added a more detailed description of the study protocols to the Materials and methods to make the experimental designs more clear and to address reviewers’ comments. These include clarifying that we had used ultracentrifugation in the preparation of peptide solutions for cross-linking studies (this addresses one of reviewer 2’s comments). In addition, we expanded the description of several experiments in the main text and the figure legends to clarify the studies, including the rationale for using a 1 h peptide incubation time on cells in oxidative stress studies and a 5 h incubation time on cells in β-cell viability studies.

Reviewer #1:

Overall, this work is on an important structural question related to IAPP toxicity in T2D. A number of spectroscopic and biochemical tools together provide some interesting information about the structural, oligomeric and toxicity states of the system as it progresses from monomer to amyloid. The effects of the anti-amyloid compounds are also important. The paper is systematically laid out. These strengths would make the work of interest to the readers of eLife. However, the work has some issues that need to be addressed. Therefore, I feel the paper is not currently at a level meriting publication in eLife.

Most importantly, the structural information provided by the work is somewhat limited. Little specific information about what structure is in which oligomeric species is obtained. While cyanoPhe and ANS experiments are good, they again do not provide specific information. The 2D IR studies are most informative, but it is hard to correlate them with specific species.

We agree that more structural information about each oligomeric species would improve the understanding of toxic and non-toxic oligomers. Unfortunately, a tradeoff is involved as the highest resolution biophysical techniques require using higher peptide concentrations that are not compatible with or comparable to our biological assays. None-the-less, the data presented in the manuscript reveals new information about h-IAPP toxic oligomers and shows that they share some properties with toxic oligomers produced by other amyloidogenic proteins, but also exhibit important differences. The observation that non-toxic variants of IAPP also oligomerize decouples general aggregation from amyloid formation; we believe this is an important result, as is the demonstration that some inhibitors of amyloid formation prolong toxicity. To further assess the properties of the different h-IAPP species produced over the course of amyloid formation, we have conducted additional dye binding experiments using bis-ANS and Nile Red. We have also expanded our discussion of the 2D IR data to describe the structural implications of those studies and to place our work in broader context.

Furthermore, the authors should provide more information about the crosslinking studies (below).

With the crosslinking studies, the control with HP35 is not quite convincing. Additional transient non-specific interactions between monomers for IAPP might also give rise to some the observed crosslinking differences. The non-toxic mutant studies may also provide some information on this aspect of the study. Authors please discuss this aspect and what they characterize as oligomeric species in more detail.

The reviewer brings up an important point; the issue of non-specific aggregation is important and has been addressed by Teplow and co-workers in their initial pioneering papers on photo-chemical cross-linking of amyloidogenic proteins (J. Biol. Chem. 2001, 276, 35176-35184; now cited in the revised manuscript). We have used their approach to show that the observed distribution of oligomers is different from that expected for nonspecific cross-linking of diffusing monomers. Please see reply to reviewer-2 comments 4 & 5 for details. We also conducted additional control experiments to verify that the observed distribution of oligomers is robust (please see reply to reviewer 2 comments 4 & 5 for details). Our data is also compatible with ion mobility mass spectroscopy studies that observed a distribution of monomers to hexamers for both h-IAPP and for r-IAPP (J. Am. Chem. Soc. 2014, 136, 660-70; cited in the revised manuscript). The suggestion to compare the non-toxic mutants with h-IAPP is interesting, however we have shown that non-toxic species aggregate and their distribution of oligomers is also very different from what is expected for non-specific “random” cross-linking. Thus, the non-toxic species studied here cannot be used to distinguish non-specific cross-linking. In fact, we believe one of the interesting aspects of our study is the observation that both toxic and non-toxic forms of IAPP oligomerize. We agree that the HP35* model is not perfect, but it does support our conclusions. We expanded our description of this protein to point out that it was designed to include the same photochemically reactive groups as h-IAPP.

How can the authors confirm that lower populations of larger oligomers (low intensity smear on gel) are not the toxic species?

We thank the reviewer for pointing this out. We repeated the cross-linking studies and the “smearing” of the gels is much less noticeable. Thus we believe part of the original smearing arose from technical staining artifacts. We also conducted new control experiments in which we examined gels of oligomeric distributions produced with different irradiation times (Figure 4—figure supplement 2). The results are described in detail in our reply to reviewer 2. The distribution is robust even for the shortest irradiation time, but the “smearing” is not detected. As mentioned above, the observation of monomers through hexamers is consistent with the mass spectroscopy studies.

Related to the above point, toxic species may involve rearrangements upon interaction with cell environments and may not be the starting species that was characterized biophysically. However, this is a general problem with many such correlations in the field and the current results are still interesting.

We agree with the reviewer’s comment. This is a general issue in the field. Our experimental design seeks to minimize this by using only a very modest dilution of the oligomeric species on the cells (only a 30% reduction in concentration) and our control experiments show that dilution into β-cell/islet culture medium (supplemented RPMI at 37^o^C) does not change the kinetics of amyloid formation.

The dilution experiment in Figure 2—figure supplement 1 seems to indicate that some of the smaller species are stable over time (it should be noted that the larger species smear is reduced by dilution) – error bars are missing for the orange bars.

Based upon the reviewer’s comments we repeated the cross-linking experiments in order to obtain sufficient data to include error bars (now presented in Figure 2—figure supplement 1 in the revised manuscript). The new data are fully consistent with our original studies. The reviewer is correct that the low order oligomers appear to be stable to the 30% dilution conducted here. We have added text about the stability of the oligomers in the revised manuscript and the possible structural implications of this observation (please also see response to reviewer 2 comments for details).

That dilution does not alter the kinetics of aggregation and needs more clarification – what does this imply for the mechanism since it is not a unimolecular process?

The kinetics of h-IAPP amyloid formation are only weakly dependent upon concentration as has been observed for a number of amyloidogenic proteins. The mechanism of h-IAPP amyloid formation is not known, but a number of amyloidogenic proteins have been shown to follow a weak power law dependence of lag time vs concentration with an exponent of -0.5 (Science, 2009, 326, 1533-1537 and Proc. Natl. Acad. Sci USA, 2008, **105,** 8926-8931). The result is expected for cases where secondary nucleation via fibril breakage is a key event. This mechanism predicts a modest lengthening of the lag time when h-IAPP is diluted from 40 µM to 28 µM (as is the case here). The experiments conducted here involved dilution from buffer into cell culture medium (supplemented RPMI at 37^o^C). The rate of h-IAPP amyloid formation is sensitive to ionic strength and to the choice of anion, even for low ionic strengths (Biochemistry 2012 51, 8478-8490). Thus, the small enhancement in lag time caused by dilution would be at least partially canceled out by the moderate change in ionic strength upon dilution into supplemented RPMI. Furthermore, the kinetics of h-IAPP amyloid formation are temperature dependent; a modest increase in temperature slightly decreases the length of the lag phase. We believe that the three small effects (dilution, modest change in ionic strength and modest change in temperature) conspire to lead to only a small change in lag time. We describe this in the caption to Figure 2—figure supplement 2 and in the first paragraph of the subsection “Toxic h-IAPP species are transient, pre-amyloid lag phase intermediates that upregulate oxidative stress, inflammation and apoptosis”. There is another, recently proposed model of h-IAPP amyloid formation that can also rationalize the lack of a significant concentration dependence of the lag time. This model hypothesizes that the lag time is controlled by a significant structural rearrangement within an oligomeric intermediate that involves crossing a high free energy barrier (Proc. Natl. Acad. Sci. USA 2013,110, 19285-90; cited in the revised manuscript]. In this model the energetics of crossing the barrier plays a significant role in dictating the lag time and the proposed rearrangement does not require a change in the association state of the intermediate. The stability of the putative intermediate is not known, but it may be that it is high enough that the small dilution used here does not significantly affect its production. We describe this model in the revised manuscript.

Reviewer #2:

This is an interesting paper describing biophysical characterization of h-IAPP oligomers. This reviewer has many questions that should be addressed prior to publication.

1) Toxicity is shown in vitro to β-cells at a concentration of 14 microM. Also, the peptide is dissolved at slightly higher concentration, and then the kinetics of the process is measured. This would appear to be an artificial system, and it is not clear how much of an extrapolation is needed to approximate the in vivo situation.

The reviewer brings up a point which is central to essentially every biophysical study of amyloid induced toxicity; namely that it is very difficult to match conditions in a controlledin vitroenvironment with those in vivo. Thus a tradeoff is required between conditions required to obtain structural/biophysical information and conditions that mimic the situation in vivo. Fortunately, this is much less of an issue for h-IAPP than for many other polypeptides. IAPP is stored in the insulin secretory granule where it is found in the halo region of the granule at concentrations estimated to be at or above 500 µM to 5 mM. Thus, the concentration of IAPP in the granule is actually higher than used in these studies. We added text to the Introduction pointing out the concentration of h-IAPP in the β-cell secretory granules. The local concentration of h-IAPP will also be high immediately upon release from the granule into the extracellular space within the islet. Dilution into β-cell/islet culture to 14 µM is as close a simulation of the in vivo situation, as is possible for an in vitro condition which is compatible with biophysical studies. Although not always appreciated, it is also worth noting that the levels of h-IAPP in transgenic mouse models of islet amyloidosis can be significantly higher than found in non-transgenic organisms since many additional copies of the transgene are present in some models and the transgene is often under control of strong promoters.

What concentration of h-IAPP is encountered in the relevant tissue? It would appear that at steady state in an animal the concentration of oligomers would be extremely low because, as the authors point out, they convert quickly to fibrils that are not toxic? Are the oligomers toxic to β-cells in tissue? Some discussion of how far this is from a natural system is warranted, particularly focusing on the effects of: 1) steady state production in vivo versus the artificial initiation in the present work; 2) are the high concentrations created in this paper relevant to what is seen in vivo; 3) would these be toxic if tested in isolated tissue or an animal model? Clearly, it is not possible to address all these issues experimentally, but the authors should clearly state the extent of the limitations of their system.

As noted above, IAPP is stored in the insulin secretory granules at concentrations between hundreds of micromoles to millimolar. As the reviewer notes, the steady state in vivo concentration of oligomers is not known for h-IAPP (or indeed for Aβ and other important amyloidogenic proteins), but, as outlined above, this may be less of an issue for h-IAPP given that the polypeptide is stored at high concentration. To address the reviewer’s important question about whether our toxic h-IAPP intermediates are toxic in isolated animal tissue, we conducted additional experiments using pancreatic islets which we isolated and hand purified from wild-type mice. The health and integrity of these organelles was confirmed via immunohistochemistry/immunofluorescence and light microscopy prior to ex vivo islet viability assays. The results show that the oligomeric h-IAPP lag phase intermediates are toxic to cells in tissue, consistent with our in vitro cellular studies. We added the new data to Figure 2 and have expanded the text in the Results and Discussion sections of the revised manuscript. We thank the reviewer for suggesting this experiment.

Experimental Questions:

1) The authors use CD to characterize the oligomeric species. Large light-scattering species are removed by centrifugation to enable the analysis. They state that the band appears similar to the α-helix, but the data are recorded in raw mdeg rather than mean residue ellipticity. If the sample is indeed helical, then one would expect a very strong mean residue ellipticity. This conversion requires knowledge of the concentration; it is possible to determine it to reasonable precision from the amide absorption (if it is possible to measure a CD spectrum it is also possible to measure an absorption spectrum, see Scopes Anal. Biochem. 59, 277, 1974 for the method). The authors should comment on whether the CD spectrum indeed has the correct intensity for an α-helix. If it is some small fraction of the expected value then it would follow that the similarity to an α-helix is not meaningful, because partially helical spectra would have a different shape.

We agree with the reviewer's comment concerning the CD and have repeated the experiments (they are very reproducible), and have replotted the CD data in the main figures as mean residue ellipticity. The results are consistent with partial helical structure in the ensemble of low order oligomers populated in the lag phase at time points of toxicity. Accurate estimates of the helical content of globular proteins can be estimated from deconvolved CD spectra. This is not likely to be as informative for the sorts of oligomers under investigation here. NMR studies of soluble analogs of IAPP at equilibrium have revealed transient fluctuating helical structure in portions of the molecule (Protein Sci. 2007, 116, 110-117). Short segments of the chain that adopt helical structure in a fluctuating ensemble may not give rise to the full rotational strength expected for a longer helix, and thus can contribute reduced intensity at 222 nm (Biopolymers 1999, 31, 569-586)

2) "Two dimensional infrared (2DIR) studies, described below, indicate that the level of β-structure is modest." Clearly, the amount of very regular β-structure (that would give rise to strong excitonic coupling) is very low from this experiment. However, it is not clear that there is sufficient concentration of protein to rule out less repetitive and partially hydrated β-structure as would be seen in globular water-soluble proteins. The authors have extensive experience in this area, so I don't doubt their conclusions. However, they should include comparisons to other literature studies from their or other groups to support this statement. I would suggest being careful with language, if the authors intend to simply rule out the possibility of the type of β-structure seen in amyloids vs. the less regular (but not necessarily disordered) β-structure seen in globular proteins.

The reviewer’s comment is well taken. We were referring to the sort of β-sheet structure that has been observed in amyloids or in β-sheet rich pre-amyloid oligomers. We have edited the text to make this clear and we have added another reference to our own work on the sensitivity of 2D IR to β-sheet structure. However, the experimental spectra of the oligomers would not arise from extensive hydrated β-sheet structure as observed in many globular proteins.

3) The use of cross-linking to probe oligomer stoichiometry is lacking in appropriate controls. Firstly, according to the experimental procedures the cross-linking is performed on the peptide solution without centrifugation. This means that the great majority of what is submitted to crosslinking is in the form of a large insoluble aggregate for h-IAPP but not for r-IAPP (earlier in the paper the authors state: "Samples of toxic h-IAPP intermediates and amyloid fibrils were pelleted at 20,000 g for 20 min and the soluble peptide remaining in the supernatant was measured…. At least 88% of h-IAPP is pelleted in the sample of fibrils, even at these low g-forces, while 94% of the peptide in the sample of toxic lag phase intermediates remains in the supernatant"). It is unclear why the samples are not centrifuged.

We thank the reviewer for pointing this out. Our initial experimental conditions did indeed include ultracentrifugation of the peptide solutions at 20,000 *g* for 20 min prior to cross-linking, but we neglected to mention this in the Methods section. We have revised the Materials and methods section and have edited the figure captions to make this clear. We have also repeated our cross-linking studies on different batches of newly synthesized peptides and obtain similar results.

4) The natural control for this experiment is to measure the extent of crosslinking near t=0, at the maximum toxicity time for the experimental conditions, and at long time after conversion to fibrils. The data at long and short time are entirely missing, and the paper should not be published without them, as the experiments would otherwise be misleading.

The reviewer's comments are very well taken, and we have carried out the requested additional cross-linking studies at “time-zero” after initiation of amyloid formation. We also conducted crosslinking studies after amyloid fibril formation is complete. The new experiments are described in the Results section where we present the other cross-linking data and in two new supplementary figures. Although not requested we conducted additional studies to verify that the observed distribution of oligomers is not an artifact of the irradiation time selected.

We measured the distribution of oligomers populated at 10 minutes. Oligomers ranging from monomers to hexamers are populated within that time frame and the relative populations are similar to those detected later in the lag phase (Figure 4, Figure 4—figure supplement 3). The data are consistent with independent ion mobility mass spectroscopy studies that report that a distribution of h-IAPP from monomers to hexamers is formed within 2 minutes of initiating h-IAPP amyloid formation, and that the distribution is present later in the lag phase [J. Am. Chem. Soc. 2014, 136, 660-70; cited in the revised manuscript]. The rapid formation of oligomers and their persistence through the lag phase is consistent with recently proposed models of h-IAPP amyloid formation, which posits that the lag phase could be controlled by a significant structural rearrangement within an oligomeric nucleus that involves crossing a high free energy barrier [Proc. Natl. Acad. Sci. USA. 2013, 110, 19285-90; cited in the revised manuscript].

The pattern of cross-linking observed for h-IAPP lag phase species is also very different than what is observed if pre-formed amyloid fibrils are cross-linked. h-IAPP was allowed to form fibrils and the samples centrifuged. No h-IAPP cross-linked oligomers were detected in the supernatant. Re-solubilization of the cross-linked fibrils revealed that the dominate species were monomers with some dimer present (Figure 4—figure supplement 5). Additional control studies show that the observed distribution is not an artifact of the irradiation time used for photochemical cross-linking (Figure 4—figure supplement 2).

5) Moreover, the authors need to factor in the cross-linking efficiency. They should provide simulations of the distribution of species expected for an infinite chain of interacting monomers with a probability of crosslinking ranging from 0.7 to 0.95 so the reader can see the sensitivity of their conclusions to the cross-linking efficiency.

We thank the reviewer for raising this important issue. Teplow and coworkers have addressed this point in their pioneering initial papers on photochemical cross-linking studies of amyloid formation (J. Biol. Chem. 2001, 276, 35176-35184; now cited in the manuscript). The analysis of Teplow and co-workers considered the case of spherical monomers which interact by diffusion with random elastic collisions. For low efficiency cross-linking the model predicts that the most populated species is the monomer and an approximately exponential decrease in intensity of high order species is predicted. Medium efficiency cross-linking still leads to the monomer being the dominate species, but to a shallower exponential decay in the relative populations. Both cases clearly differ from that observed for h-IAPP. High efficiently cross-linking is predicted to lead to further consumption of monomers and the maximum shifting to dimers with the predicted monomer and dimer populations being higher than the predicted trimer, tetramer and pentamers populations. Again, this distribution is fundamentally different from that observed for IAPP lag phase species, where trimers are the most highly populated species and the population of pentamers is comparable to the population of monomer. We now discuss these issues when we present the cross-linking data.

6) Finally, the authors argue that the oligomers are held together by highly flexible interactions that would be expected to equilibrate rapidly. However, they show experimentally that there is essentially no change in the cross-linking pattern when oligomers are diluted and allowed to equilibrate. One would expect that this would lead to extensive dissociation of the oligomers (note that the stability of an n-mer depends on the nth value of the monomer concentration). Thus, one might expect rapid dissociation from even a modest dilution. It is surprising that the authors fail to discuss this apparent paradox.

We were not sufficiently rigorous in our wording and thank the reviewer for pointing this out. The apparent stability of the oligomers is not known. The oligomers observed in our cross-linking studies could be stabilized by intermolecular hydrogen bonding and packing of small segments of the chain provided this does not lead to sequestering of the aromatic residues from solvent, and the development of hydrophobic patches sufficient to bind ANS, bis-ANS or Nile Red (dye-binding studies presented in Figure 7 and Figure 7—figure supplement 2 in the revised manuscript). Along these lines, real time isotope edited 2D IR studies conducted at higher concentrations (the isotope edited version of the experiment provides specific structural information, but is currently impossible to conduct at the concentrations used in the present studies because of hardware limitations) indicate that an intermolecular β-sheet may be formed during amyloid assembly which involves the FGAIL region. Our data is fully compatible with this model. We have edited the text to be clear on this point. Thus, the modest dilution in our studies (only 30% from 40 µM to 28 µM) may not be sufficient to lead to a level of dissociation detectable via cross-linking. It is also not known in the present case if n-mer production strictly involves a monomer to n-mer pre-equilibrium or if, for example, hexamers can also form via a pre-equilibrium involving, dimers and tetramers and tetramer formation by binding of dimers to dimers, etc.